# Immunological and clinicopathological features predict HER2-positive breast cancer prognosis in the neoadjuvant NeoALTTO and CALGB 40601 randomized trials

Mattia Rediti [1], Aranzazu Fernandez-Martinez[2], David Venet[1], Françoise Rothé[1], Katherine A. Hoadley [2], Joel S. Parker [2], Baljit Singh[3], Jordan D. Campbell[4], Karla V. Ballman[5], David W. Hillman [4], Eric P. Winer[6], Sarra El-Abed[7], Martine Piccart [8], Serena Di Cosimo[9], William Fraser Symmans [10], Ian E. Krop[6], Roberto Salgado [11,12], Sherene Loi[12], Lajos Pusztai[13], Charles M. Perou [2], Lisa A. Carey [14] & Christos Sotiriou [1] ✉

The identification of prognostic markers in patients receiving neoadjuvant therapy is crucial for treatment optimization in HER2-positive breast cancer, with the immune microenvironment being a key factor. Here, we investigate the complexity of B and T cell receptor (BCR and TCR) repertoires in the context of two phase III trials, NeoALTTO and CALGB 40601, evaluating neoadjuvant paclitaxel with trastuzumab and/or lapatinib in women with HER2-positive breast cancer. BCR features, particularly the number of reads and clones, evenness and Gini index, are heterogeneous according to hormone receptor status and PAM50 subtypes. Moreover, BCR measures describing clonal expansion, namely evenness and Gini index, are independent prognostic factors. We present a model developed in NeoALTTO and validated in CALGB 40601 that can predict event-free survival (EFS) by integrating hormone receptor and clinical nodal status, breast pathological complete response (pCR), stromal tumor-infiltrating lymphocyte levels (%) and BCR repertoire evenness. A prognostic score derived from the model and including those variables, HER2-EveNT, allows the identification of patients with 5-year EFS > 90%, and, in those not achieving pCR, of a subgroup of immune-enriched tumors with an excellent outcome despite residual disease.

Neoadjuvant treatment escalation approaches with dual anti-human epidermal growth factor receptor 2 (HER2) blockade have been proven effective in early-stage HER2-positive breast cancer, leading to an increase in pathological complete response (pCR) rates[1,2]. In the neoadjuvant lapatinib and/or trastuzumab treatment optimization (NeoALTTO) trial, dual anti-HER2 blockade with the tyrosine kinase inhibitor lapatinib and the monoclonal antibody trastuzumab

improved pCR rates in comparison to each targeted agent alone, and a survival benefit was demonstrated for patients achieving pCR[1,3,4]. Moreover, the cancer and leukemia group B (CALGB) 40601 trial reported a significant survival benefit for dual HER2-targeting with lapatinib and trastuzumab compared to the single anti-HER2 agents[5,6].

HER2-positive breast cancer is considered an immunogenic tumor, and the host immune response in this breast cancer subtype

has been extensively explored due to its known role in modulating the activity of anti-HER2 agents [e.g., through antibody-dependent cellular cytotoxicity (ADCC)][7,8]. Therefore, understanding the immune micro-environment in HER2-positive breast cancer represents a key element for biomarker research in this disease. In the NeoALTTO trial, higher levels (expressed in %) of stromal tumor-infiltrating lymphocytes (TILs) predicted improved responsiveness and prognosis[9], while immune gene signatures[10] and usage of specific T cell receptor (TCR) $\beta$ chain variable genes[11] were positively associated with response in the combination arm. A prominent role of immune gene signatures (particularly related to B cell response) in predicting both pCR and prognosis has also been described in the CALGB 40601 study, even outperforming TILs in multivariable analyses[5,6,12], opening questions regarding the need to integrate morphology with a more specific characterization of immune cell subtypes. An association with TIL frequencies has also been described for PAM50 subtypes[13,14] and hormone receptor status[9].

Of note, increased diversity and clonal expansion of B cell receptor (BCR) and TCR repertoires, which can be described with different metrics (Supplementary Fig. 1) are characteristics of the immune response[15,16]. Interestingly, several studies have recently focused their efforts on characterizing the role of B cells in different tumor types, including breast cancer[16–26]. These works have linked B cell infiltration and clonal diversity with improved outcomes as well as responsiveness to immunotherapy, and have depicted the role of B cells in orchestrating the immune response in tumors.

Moreover, the achievement of pCR after neoadjuvant treatments is a robust prognostic indicator[27,28]. However, many patients with residual disease (RD) never experience recurrence, whereas disease relapse can still be observed in a subgroup of patients achieving pCR.

In this retrospective study, we further dissect the heterogeneity of HER2-positive breast cancer by evaluating the characteristics of the BCR and TCR repertoires in the context of two phase III clinical trials, namely NeoALTTO and CALGB 40601, evaluating neoadjuvant paclitaxel with trastuzumab and/or lapatinib in women with HER2-positive breast cancer. Moreover, we explore the impact of BCR and TCR repertoires and diversity measures on treatment response and long-term clinical outcomes. We present a prognostic model integrating baseline immunological and clinical features, as well as treatment response, aimed at identifying distinct prognostic groups in HER2-positive breast cancer patients treated with neoadjuvant therapies. The prognostic stratification of patients and the biological characterization of determinants of prognosis can open the avenue for the implementation of optimized neoadjuvant/post-operative treatment strategies.

## Results

### Characteristics of the RNA sequencing cohorts in the NeoALTTO and CALGB 40601 trials

The characteristics of the population with baseline pre-treatment RNA sequencing data in NeoALTTO ($N = 254$) and CALGB 40601 ($N = 264$) have been previously described[6,10]. When comparing the two cohorts, relevant differences were noted (Table 1). In particular, the CALGB 40601 cohort included a population characterized by more favorable clinical features (fewer $\geq$ cT3 tumors, more patients with clinically negative lymph nodes, higher TIL levels) compared to the NeoALTTO one. Higher pCR rates were also observed in CALGB 40601 compared to NeoALTTO (Table 1). Median follow-up for the NeoALTTO RNA sequencing cohort was 6.7 years, while for CALGB 40601, it was 9.1 years. Consort diagrams showing the number of patients included for the subsequent analyses are available in Supplementary Fig. 2.

### BCR and TCR repertoires in the NeoALTTO and CALGB 40601 trials

Aiming at investigating the complexity of the immune response in HER2-positive breast cancer, we explored the diversity of BCR and TCR

**Table 1 | Baseline patients' characteristics and pathological complete response rates of the NeoALTTO and CALGB 40601 RNA sequencing cohorts**

| | NeoALTTO (N = 254) | CALGB 40601 (N = 264) | P value |
|---|---|---|---|
| Age | | | 0.735[a] |
| - Median (Q1, Q3) | 49.0 (41.2, 56.8) | 49.0 (41, 56) | |
| Racial or ethnic group | | | $1.11 \times 10^{-12b}$ |
| - Asian | 69 (27.2%) | 16 (6.1%) | |
| - Black | 4 (1.6%) | 21 (8.0%) | |
| - White | 158 (62.2%) | 213 (80.7%) | |
| - Other | 23 (9.1%) | 14 (5.3%) | |
| Tumor size | | | $1.75 \times 10^{-9b}$ |
| - Not available | 0 | 16 | |
| - cT1 | 0 (0.0%) | 24 (9.7%) | |
| - cT2 | 151 (59.4%) | 161 (64.9%) | |
| - $\geq$ cT3 | 103 (40.6%) | 63 (25.4%) | |
| Nodal stage | | | $9.05 \times 10^{-8b}$ |
| - cN0 | 67 (26.4%) | 113 (42.8%) | |
| - cN1 | 149 (58.7%) | 110 (41.7%) | |
| - cN2 | 24 (9.4%) | 23 (8.7%) | |
| - cN3 | 13 (5.1%) | 3 (1.1%) | |
| - cNx | 1 (0.4%) | 15 (5.7%) | |
| HR status | | | 0.331[b] |
| - Negative | 117 (46.1%) | 110 (41.7%) | |
| - Positive | 137 (53.9%) | 154 (58.3%) | |
| ER status | | | 0.113[b] |
| - Negative | 128 (50.4%) | 114 (43.2%) | |
| - Positive | 126 (49.6%) | 150 (56.8%) | |
| pCR breast (ypT0/is) | | | 0.007[b] |
| - No | 166 (65.4%) | 141 (53.4%) | |
| - Yes | 88 (34.6%) | 123 (46.6%) | |
| pCR breast + axilla (ypT0/is ypN0) | | | 0.008[b] |
| - Not available | 10 | 0 | |
| - No | 169 (69.3%) | 152 (57.6%) | |
| - Yes | 75 (30.7%) | 112 (42.4%) | |
| Treatment Arm | | | 0.003[b] |
| - Lapatinib (L) | 89 (35.0%) | 57 (21.6%) | |
| - Trastuzumab (T) | 79 (31.1%) | 104 (39.4%) | |
| - T + L | 86 (33.9%) | 103 (39.0%) | |
| TILs (%)[c] | | | $6.79 \times 10^{-10a}$ |
| - Median (Q1, Q3) | 12.5 (5, 32.5) | 20 (13.125, 45) | |
| PAM50 | | | 0.302[b] |
| - HER2-Enriched | 151 (59.4%) | 146 (55.3%) | |
| - Basal-like | 21 (8.3%) | 22 (8.3%) | |
| - Luminal B | 41 (16.1%) | 35 (13.3%) | |
| - Luminal A | 22 (8.7%) | 28 (10.6%) | |
| - Normal-like | 19 (7.5%) | 33 (12.5%) | |

P values are two-sided.
BCR B cell receptor, ER estrogen receptor, HR hormone receptor, pCR pathological complete response, Q1 quartile 1, Q3 quartile 3, TILs tumor-infiltrating lymphocytes.
[a]Wilcoxon rank sum test.
[b]Fisher's Exact Test.
[c]TILs data were available for 233 patients in NeoALTTO and 230 patients in CALGB 40601.

repertoires in pre-treatment baseline tumor samples. The MiXCR tool[29,30] was used to identify BCR and TCR clones from RNA sequencing data, with some differences noted between NeoALTTO and CALGB 40601.

In NeoALTTO, all samples had at least one read mapping to BCR, while one sample did not have any read mapping to TCR. In the CALGB

40601 study, all samples showed reads mapping to both BCR and TCR. As the library size can impact the number of BCR/TCR reads, for the subsequent analyses we normalized the number of BCR/TCR reads by the total number of reads mapping to genes in each sample. The number of reads mapping to TCR was significantly lower than BCR in both studies, and the number of normalized reads and clones were significantly higher in NeoALTTO compared to CALGB 40601 (Supplementary data 1). In this regard, differences in the RNA sequencing between the two studies, namely the read length [50-base pairs (bp) in CALGB 40601 and 100-bp in NeoALTTO] have to be considered. Indeed, while different read lengths can be used in MiXCR, 100-bp read libraries allow detection of higher numbers of clonotypes compared to shorter read lengths[31], potentially leading to the identification of a lower number of clones in CALGB 40601. This was further confirmed when testing MiXCR in NeoALTTO after trimming reads at 50-bp (details in METHODS), comparing the read and clone counts obtained with the original NeoALTTO data used for the analyses (Supplementary data 1).

The proportions of reads mapping to the different immunoglobulin and TCR chains are shown in Supplementary Fig. 3a–d. In both studies, the majority of the reads were generated from the κ light chain for BCR repertoires, and from both the $\alpha$ and the $\beta$ chain for TCR. To describe the characteristics of the BCR/TCR repertoires, we computed diversity indices derived from economics[32] and ecology studies[33,34] previously applied to BCR/TCR repertoires characterization[17,35], including Gini index, Gini-Simpson index and species evenness (defined in the METHODS). The proportion of the most frequent (top) and the second clone, and the length in nucleotides of the complementarity determining region 3 (CDR3) were also calculated (Supplementary data 2, 3). Illustrative examples are depicted in Supplementary Fig. 1. Correlations between measures computed on the single-chain types and the total BCR/TCR measures (calculated considering all reads mapping to any BCR or TCR gene) are shown in Supplementary Fig. 4 for NeoALTTO and Supplementary Fig. 5 for CALGB 40601 (values reported in Supplementary data 4, 5). Except for the CDR3 length, which differed in the various chains (Supplementary data 1), and TCR gamma and delta chains, presenting a low number of reads, we observed moderate to high correlations (rho > 0.50) with the total BCR/TCR metrics for the majority of the single-chain features in both trials.

From here on, we refer to BCR/TCR repertoire characteristics as the global metrics calculated on all reads mapping to chains forming either the BCR or TCR, with clones defined separately for each chain as detailed in the METHODS, representing the whole clonal repertoire.

The correlation among different BCR/TCR measures is also shown in Supplementary Figs. 4 and 5. BCR/TCR read counts were positively correlated to the number of clones and Gini index, while inverse correlations were noted with evenness, top and second top clone proportions, showing that the presence of a higher number of reads mapping to BCR/TCR was also associated to the detection of a more diverse repertoire.

Isotypes (IgG, IgA, IgM, IgD, IgE) from BCR heavy chains were also computed (details in METHODS). In both studies, the majority of the BCR heavy chain reads were from IgG, followed by IgA and IgM (Supplementary Fig 3e–f). As shown in Supplementary Fig. 6, positive correlations (the highest being for IgG) were observed between selected isotypes diversity measures and the corresponding global BCR measures (values reported in Supplementary data 6, 7).

### BCR and TCR repertoires are heterogeneous according to hormone receptor status and PAM50 subtypes, and correlate with TIL levels

We next explored the association of BCR and TCR repertoires with hormone receptor status, PAM50 subtypes, and TIL levels, scored as % of the intratumoral stroma area following the International TILs

Working Group guidelines[36], in order to depict the heterogeneity of the immune response within HER2-positive breast cancer and evaluate whether BCR/TCR measures could add additional information to TILs.

As shown in Fig. 1 and Supplementary Figs. 7 and 8, we observed differences between hormone receptor-negative (HR-) and hormone receptor-positive (HR+) tumors (full results in Supplementary data 8). In particular, HR- tumors had significantly higher BCR read counts and number of clones compared to HR+ tumors in both NeoALTTO and CALGB 40601, suggesting higher levels of B cell infiltration in line with a higher immunogenicity attributed to HR- tumors[8].

We then compared BCR and TCR measures according to PAM50 subtypes. For this purpose, in order to have uniformity in the PAM50 analyses, we adopted in NeoALTTO the method described in Fernandez-Martinez et al. [6]. NeoALTTO and CALGB 40601 presented similar populations in terms of PAM50 classification (Table 1, Supplementary data 9). The proportion of the PAM50 subtypes was also similar in HR+ and HR- tumors in the two trials (Supplementary Fig 9a–f). Rates of pCR (ypT0/is) in NeoALTTO according to the PAM50 subtypes are shown in Supplementary Fig. 9g, with HER2-enriched (HER2-E) tumors having the highest pCR rate (43.7%). Differently from what has been demonstrated in CALGB 40601[6], event-free survival (EFS) in NeoALTTO did not differ significantly among PAM50 subtypes (Supplementary Fig 9h–j), nor did HER2-E tumors have a worse prognosis compared to the other PAM50 subtypes amongst those with RD.

Of interest, we observed heterogeneity in terms of BCR repertoire across PAM50 subtypes for both trials, with different distributions (Kruskal-Wallis FDR < 0.05) for read counts, evenness, Gini index, Gini-Simpson, as well as top and second clone proportions (Fig. 1 and Supplementary Fig. 10). TCR measures were overall more homogeneous (Supplementary Fig. 11), although some differences could be noted (e.g., differences in the number of reads in both studies). Overall, HER2-E, basal-like, and luminal B tumors showed features in line with an activation of the B-cell mediated immune response and clonal expansion, as described by higher BCR Gini index and lower BCR evenness, compared to luminal A and normal-like samples. Detailed results for the comparisons are also shown in Supplementary data 10, 11.

TIL levels at baseline were available for 233 patients out of the 254 (91.73%) with pre-treatment RNA sequencing data in the NeoALTTO study and 230 out of 264 (87.12%) in the CALGB 40601 cohort. As shown in Supplementary Figs. 4, 5, a moderate positive correlation (rho = 0.33 to 0.58) with TIL levels was noted for the BCR and TCR normalized number of reads (similar to the moderate correlation between TIL levels and immune-related gene expression signatures recently reported[12]), the number of clones as well as for BCR Gini index. In contrast, a negative correlation with TIL levels (rho = −0.29 to −0.41) was noted for BCR evenness, as well as TCR top and second clone proportions. As top and second top clone proportions are also negatively correlated with the number of clones, higher values may represent populations with low levels of both immune infiltration and number of clones. Since correlations between TIL levels and BCR/TCR diversity measures were moderate, the information brought by TILs quantification and BCR/TCR characterization may be complementary, with diversity measures potentially providing a "qualitative" information in terms of immune response compared to the quantification of TILs.

### BCR/TCR repertoire metrics can predict pCR and EFS

To assess whether baseline BCR and TCR measures could predict pCR (Supplementary Figs. 12, 13) and EFS (Fig. 2), we performed univariable and multivariable analyses in the NeoALTTO and CALGB 40601 cohorts, demonstrating a potential prognostic role for BCR diversity.

When controlling for clinicopathological characteristics, higher TCR evenness and lower Gini index, top and second clone proportions

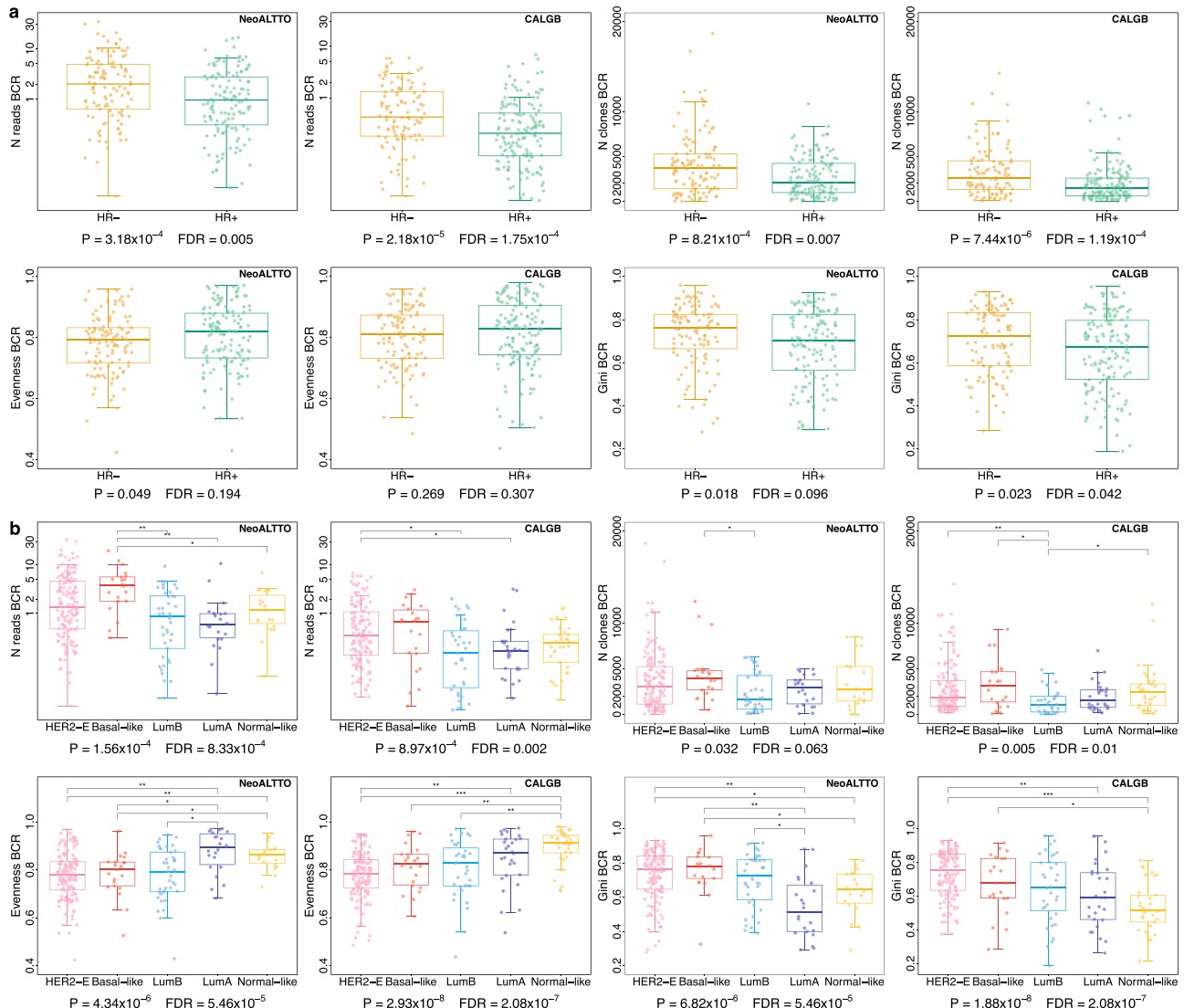

**Fig. 1 | Heterogeneity of BCR measures according to hormone receptor status and PAM50 subtypes. a** Comparisons of BCR normalized number of reads ("N reads"; represented on a log scale), number of clones ("N clones"), evenness and Gini index in HR- and HR+ HER2-positive breast cancer in NeoALTTO ($N = 254$) and CALGB 40601 ($N = 264$). Two-sided $P$ values at the bottom of the panels are from Wilcoxon rank sum test, and FDRs obtained adjusting $P$ values using Benjamini & Hochberg method. See also Supplementary Fig. 7 for the other BCR measures and Supplementary data 8 reporting $P$ values and FDRs. **b** Comparisons of BCR normalized number of reads ("N reads"; represented on a log scale), number of clones ("N clones"), evenness and Gini index in PAM50 subtypes in HER2-positive breast cancer in NeoALTTO ($N = 254$) and CALGB 40601 ($N = 264$). Two-sided $P$ values at the bottom of the panels are from Kruskal-Wallis test, while Wilcoxon rank sum test was used when comparing each group against each one of the others. FDRs were then obtained adjusting $P$ values using Benjamini & Hochberg method. See also Supplementary Fig. 10 for the other BCR measures, Supplementary data 10, 11 reporting $P$ values and FDRs. Source data are available. For Wilcoxon tests, FDRs < 0.05 are shown. In the panels: * = FDR < 0.05 and 0.01; ** = FDR < 0.01 and ≥ 0.001; *** = FDR < 0.001. The number of reads is normalized by the total number of reads mapping to the transcriptome in each sample, and multiplied by 1000. In boxplots, the boxes are defined by the upper and lower quartile; the median is shown as a bold colored horizontal line; whiskers extend to the most extreme data point which is no more than 1.5 times the interquartile range from the box. BCR B cell receptor, FDR false discovery rate, HER2-E HER2-Enriched, HR hormone receptor, LumA luminal A, LumB luminal B.

were associated with higher pCR (ypT0/is) rates in NeoALTTO, while higher BCR number of reads, number of clones, and Gini index were positively associated with pCR in CALGB 40601. When evaluating breast + axilla pCR, only BCR reads, and clone numbers in CALGB 40601 were significant after controlling for clinicopathological characteristics.

BCR/TCR features were not significantly associated with EFS at the univariable analysis in NeoALTTO and CALGB 40601 after adjusting the $P$ values for multiple testing (Fig. 2a, b). When controlling for clinicopathological characteristics, BCR evenness [hazard ratio (HR) = 1.5; FDR = 0.021] and Gini index (HR = 0.69; FDR = 0.021) were significantly associated with EFS in NeoALTTO (Fig. 2c), while BCR read

count (HR = 0.51; FDR = $1.9 \times 10^{-4}$), number of clones (HR = 0.58; FDR = 0.0013), evenness (HR = 1.7; FDR = 0.025) and Gini index (HR = 0.6; FDR = 0.015) were associated with EFS in CALGB 40601 (Fig. 2d).

Multivariable EFS analyses for immunoglobulin chains and isotypes (IgG, IgM, IgA only, due to the low proportions of IgD and IgE detected) are shown in Supplementary Fig. 14. While the directions of the association are in line with the described BCR results, focusing on single chains (e.g., heavy chain) or isotype (e.g., IgG) may provide less prognostic information compared to measures on the global BCR repertoire.

These results suggest a positive prognostic role of B cell clonal expansion, depicted by lower BCR evenness and higher Gini index.

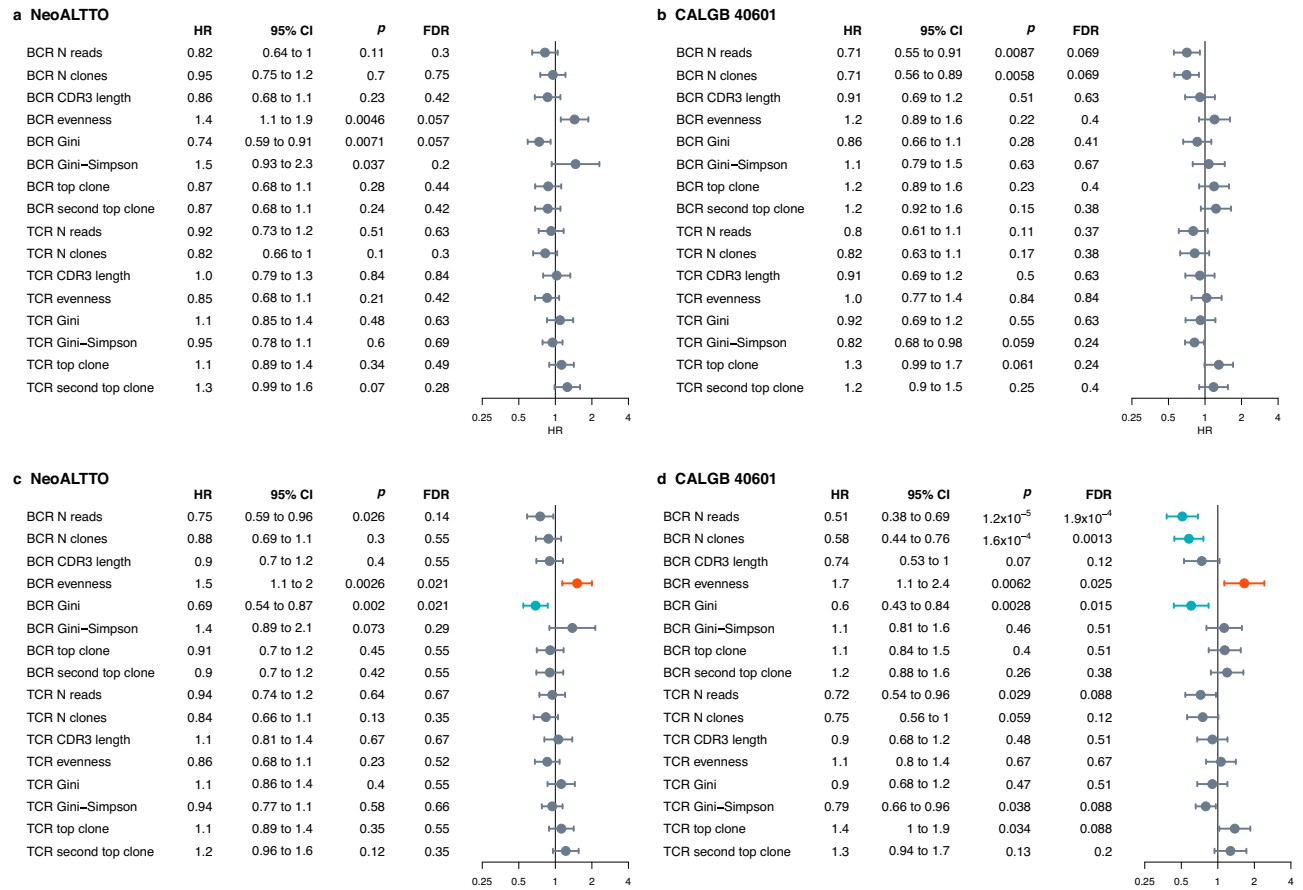

**Fig. 2 | Association of BCR and TCR measures with EFS in the NeoALTTO and CALGB 40601 cohorts. a** Forest plot for EFS in the NeoALTTO cohort, univariable analysis. **b** Forest plot for EFS in the CALGB 40601 cohort, univariable analysis. **c** Forest plot for EFS in the NeoALTTO cohort, correcting for clinicopathological parameters (age, hormone receptor status, tumor size, nodal status, PAM50 subtypes, and treatment arm). **d** Forest plot for EFS in the CALGB 40601 cohort, correcting for clinicopathological parameters (age, hormone receptor status, tumor size, nodal status, PAM50 subtypes, and treatment arm). For univariable analysis, *P* values are from likelihood ratio test. When correcting for clinicopathological characteristics, *P* values were obtained with an ANOVA on nested Cox models. *P* values are two-sided. Non-significant values (FDR > 0.05) are shown in dark gray, significant values are shown in red (HR > 1) and blue (HR < 1). Circles indicate HR, and error bars the 95% confidence interval (95% CI). Analyses were performed including patients with available data. In NeoALTTO, *N* = 254 for all BCR/TCR metrics, except TCR CDR3 length, TCR Gini, TCR Gini-Simpson, TCR top clone (*N* = 253) and TCR evenness, TCR second top clone (*N* = 251). In CALGB 40601, for all BCR/TCR measures *N* = 264 in univariable, *N* = 248 in multivariable. 95% CI 95% confidence interval, BCR B cell receptor, CDR3 complementarity-determining region 3, EFS event-free survival, FDR false discovery rate, HR hazard ratio, N reads number of normalized reads, N clones number of clones, TCR T cell receptor.

## Development and validation of an integrated prognostic model

Considering the previous results, we tested whether a model including baseline clinicopathological, immune-related features, and treatment-related information including response could predict EFS in NeoALTTO. BCR and TCR measures (details in METHODS) were included in the variable selection process, together with pCR information, clinicopathological variables and a set of gene signatures. The model was developed on a training cohort of 221 patients from the NeoALTTO trial with all included data available, using a forward step-wise approach with Akaike Information Criteria (AIC) to select the best model (Supplementary data 12).

The variables selected in the final model included BCR evenness, TIL levels, pCR status (ypT0/is), clinical nodal status and hormone receptor status (the final Cox regression model is shown in Supplementary data 12). A score, named HER2-EveNT for hormone receptor status, pCR, BCR evenness, nodal status, and TILs, was then calculated as the sum of the values assigned to each variable multiplied by the estimated coefficient derived from the Cox model (details in METHODS).

In NeoALTTO, the C-index calculated on the training cohort (*N* = 221) was 0.689, while the C-index on the group of patients with available data for the variables selected in the prognostic model

(*N* = 233, used for the subsequent analyses) was 0.6979 (score range: −3.3176 to 0.3665, median −1.0113, with a lower score being associated with better outcome).

The score was then divided into tertiles (cutoffs at −1.3763 and −0.8143), and Kaplan–Meier analysis was performed to estimate EFS in the groups identified (Supplementary Fig. 15a). Five-year EFS rates were 92% (95% CI = 86–98%), 75% (95% CI = 66–86%) and 55% (95% CI = 44–68%) for the first, second and third tertile, respectively. The second and third tertiles, presenting 5-year EFS rates < 90%, were merged into a single group, defining two final prognostic groups with good (*N* = 78) and poor (*N* = 155) prognosis (Fig. 3a). Five-year EFS rate for the poor prognosis group was 65%, with HR = 0.2 for good vs. poor prognosis (log-rank *P* = 1.21 × 10⁻⁵).

Furthermore, by separating the patients according to breast pCR (ypT0/is) status, we were able to identify subgroups of patients with distinct prognosis regardless of treatment response (Fig. 3b–c) due to the impact of the other variables in these subgroups. In particular, patients with breast pCR had 5-year EFS rates of 97% and 72% in the two prognostic groups (log-rank *P* = 0.00057; HR = 0.12). In patients not achieving pCR in the breast, 5-year EFS rates were 85% in the good prognosis group and 62% in the poor prognosis group (log-rank *P* = 0.012; HR = 0.32).

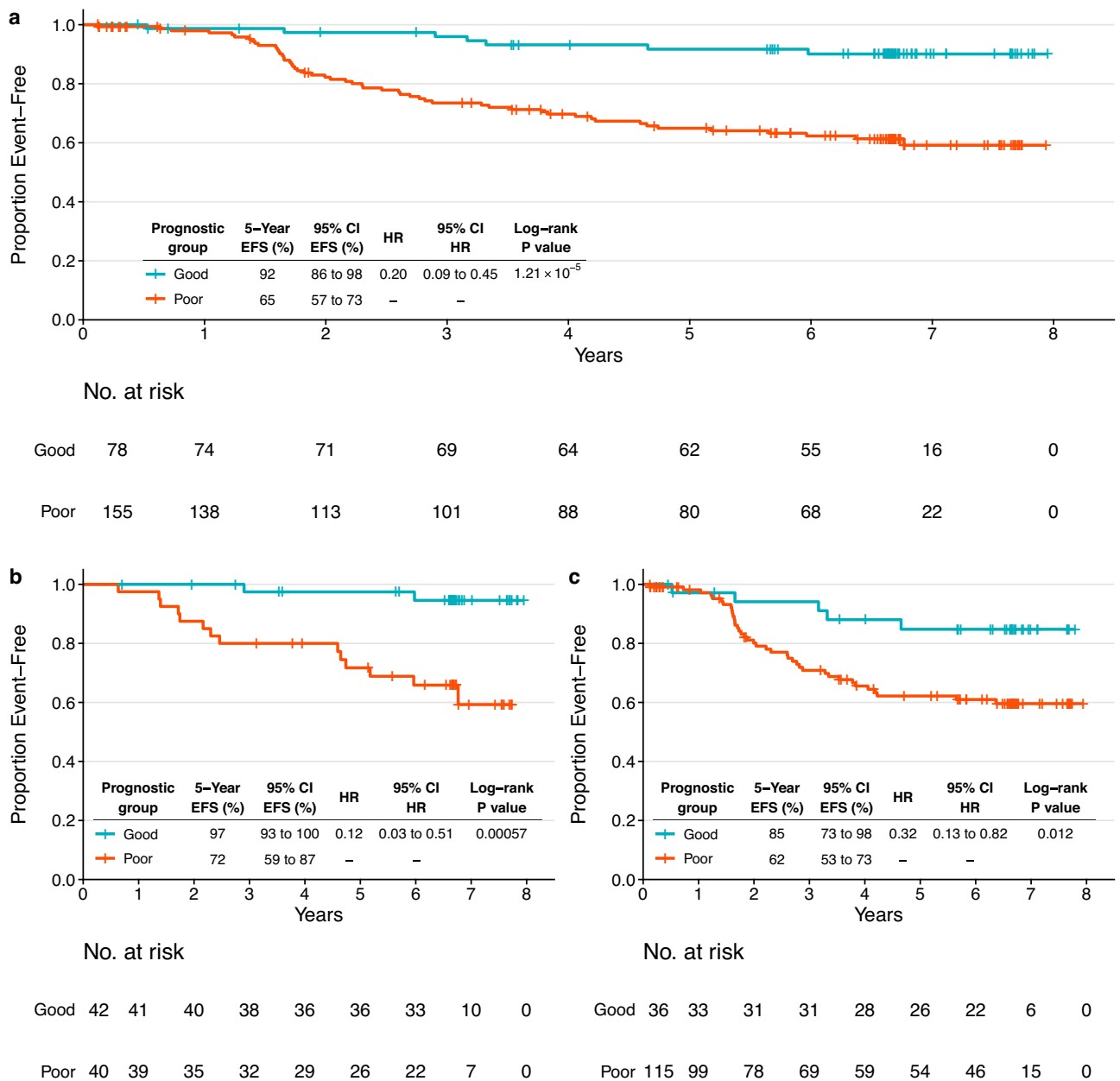

**Fig. 3 | Event-free survival outcomes based on the groups identified by the prognostic HER2-EveNT score in the NeoALTTO dataset. a** Kaplan–Meier plot showing EFS in the NeoALTTO population ($N = 233$) with information available for all variables included in the model (breast pCR, hormone receptor status, clinical nodal status, TILs, BCR evenness). **b** Kaplan–Meier plot showing EFS in the NeoALTTO subgroup with all variables in the model available and breast pCR (ypT0/is) at surgery ($N = 82$). **c** Kaplan–Meier plot showing EFS in the NeoALTTO subgroup with all variables in the model available and without pCR in the breast at surgery ($N = 151$). Patients are stratified according to low risk (good prognosis group, first tertile) and high risk (poor prognosis group, tertiles 2 and 3 combined), based on the HER2-EveNT score derived from the prognostic model. Tables show 5-year EFS rates and HRs with respective 95% CI. $P$ values are from log-rank test, HR describes the risk of an event as defined by EFS in the good prognosis group compared to the one with poor prognosis. 95% CI 95% confidence interval, BCR B cell receptor, EFS event-free survival, HR hazard ratio, pCR pathological complete response, TILs tumor-infiltrating lymphocytes.

We then proceeded to validate the prognostic model by computing the HER2-EveNT score in the CALGB 40601 set ($N = 230$) and applying the pre-defined cutoffs from NeoALTTO (Supplementary Figs. 15b, 4, details in METHODS). In CALGB 40601, 119/230 patients (51.96%) were included in the good prognosis group (score ≤ −1.3763), in line with the favorable clinical features previously described compared to NeoALTTO. The C-index in CALGB 40601 cohort was 0.6396 (score range: −3.8219 to 0.2597, median −1.42). HER2-EveNT scores are available in Supplementary data 13, 14.

Five-year EFS rates were 93% and 75% for the good and poor prognostic group, respectively (log-rank $P = 8.44 \times 10^{-5}$; HR = 0.25). The

difference in the pCR group (Fig. 4b) was not statistically significant, although a trend similar to the one observed in NeoALTTO was noted (log-rank $P = 0.19$; HR = 0.43), with 5-year RFS rates of 93% for the good prognosis and 85% for the poor prognosis groups. Indeed, the power to detect significant differences in survival among patients who achieved pCR is hindered by the good prognosis of those patients. Nevertheless, a difference of 8% in the 5-year EFS rates was observed between the two prognostic groups. Confirming the findings in NeoALTTO, groups with distinct prognosis were identified in patients not achieving breast pCR (Fig. 4c), with 5-year EFS rates of 93% and 72% for the good and poor prognosis group, respectively (log-rank $P = 0.009$; HR = 0.27).

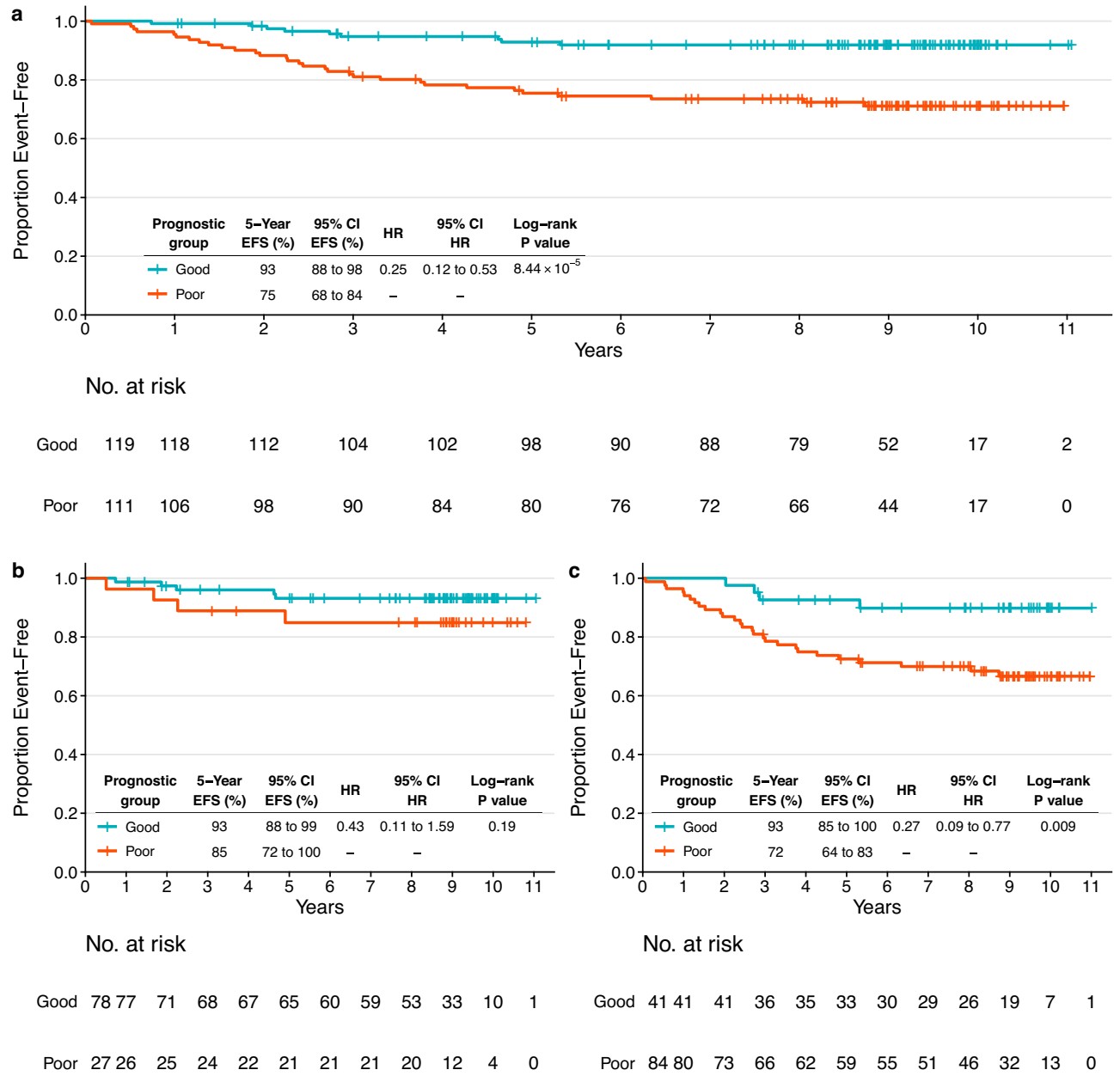

**Fig. 4 | Event-free survival outcomes based on the groups identified by the prognostic HER2-EveNT score in the CALGB 40601 independent validation dataset. a** Kaplan–Meier plot showing EFS in the CALGB 40601 population (N = 230) with information available for all variables included in the model (breast pCR, hormone receptor status, clinical nodal status, TILs, BCR evenness). **b** Kaplan–Meier plot showing EFS in the CALGB 40601 subgroup with all variables in the model available and breast pCR (ypT0/is) at surgery (N = 105). **c** Kaplan–Meier plot showing EFS in the CALGB 40601 subgroup with all variables in the model available and without pCR in the breast at surgery (N = 125). Patients are stratified according to low risk (good prognosis group) and high risk (poor prognosis), based on the HER2-EveNT score derived from the prognostic model. The cutoff to identify the prognostic groups is derived from NeoALTTO (patients with a score ≤ −1.3763 were assigned to the good prognosis group). Tables show 5-year EFS rates and HRs with respective 95% CI. P values are from log-rank test, HR describes the risk of an event as defined by EFS in the good prognosis group compared to the one with poor prognosis. 95% CI 95% confidence interval, BCR B cell receptor, EFS event-free survival, HR hazard ratio, pCR pathological complete response, TILs tumor-infiltrating lymphocytes.

In addition, similar results and differences in terms of survival were observed when dividing the NeoALTTO and CALGB 40601 cohorts according to pCR defined as ypT0/is ypN0 (Supplementary Fig. 16).

As expected, when considering the score as a continuous variable, the prognostic score was associated to EFS at the univariable analysis in NeoALTTO (HR = 2.2; 95% CI = 1.6–3; $P = 5.2 \times 10^{-8}$), as well as after adjusting for treatment arm, tumor size, PAM50 subtypes and age (HR = 2.2; 95% CI = 1.6–2.9; $P = 9.3 \times 10^{-8}$). These results were confirmed in CALGB 40601 (univariable analysis: HR = 1.6; 95% CI = 1.1–2.2;

$P = 0.0066$; multivariable analysis: HR = 1.8; 95% CI = 1.3– 2.5; $P = 7.4 \times 10^{-4}$).

Results for overall survival (OS) are shown in Supplementary Figs. 17, 18. In both studies, OS results had the same trend compared to EFS; however, in NeoALTTO a statistically significant difference was observed only in the whole population and not in the pCR/no-pCR subgroups, while in CALGB 40601 statistically significant differences were observed in the pCR/no-pCR subgroups as well.

The characteristics of the prognostic groups are shown in Supplementary Tables 1, 2. Interestingly, the proportions of patients

without pCR (ypT0/is) in the good prognosis group were 46.2% and 34.5% in the NeoALTTO and CALGB 40601 trials, respectively. There were no significant differences in terms of anti-HER2 treatment received, and the difference in EFS outcome between the two prognostic groups could be observed across all treatment arms (Supplementary Fig. 19). Furthermore, in each treatment arm the prognostic information associated to the HER2-EveNT groups in the pCR/no-pCR subsets seems to maintain the same trends observed in the whole cohort (Supplementary Fig. 20), although these results require caution in the interpretation due to the small number of patients in each one of these comparisons.

When comparing EFS in patients belonging to the good prognosis group with and without pCR (ypT0/is), we did not find significant differences neither in NeoALTTO (log-rank $P = 0.13638$; HR = 0.31; 95% CI = 0.06–1.59), nor in CALGB 40601 (log-rank $P = 0.557$; HR = 0.68; 95% CI = 0.18–2.52).

Similar variables (i.e., TILs, BCR evenness, nodal status, and estrogen receptor status) where selected when removing the pCR and treatment information from the pool of variables tested, although this version of the EFS model presented a reduced performance compared to the original one, both in NeoALTTO (C-index 0.6787 vs. 0.6979 of the original model) and CALGB 40601 (C-index 0.6029 vs. 0.6396).

Overall, a multi-modal approach integrating clinicopathological characteristics, response information and immunological features is important to predict patients' outcomes.

## Tumor microenvironment characteristics are associated with BCR/TCR repertoire metrics and the prognostic score

To explore the relationship of BCR/TCR characteristics, TILs and prognosis with the tumor microenvironment composition in the NeoALTTO and CALGB 40601 cohorts, we applied the microenvironment cell populations (MCP)-counter tool[37], which allows to estimate the abundance scores of different immune and stromal cells (Supplementary data 15, 16). Indeed, differences in the microenvironment composition were associated to BCR/TCR measures and TIL levels, as well as to the HER2-EveNT prognostic score.

We first evaluated the correlations of the abundance scores for each cell type were with the BCR/TCR measures, TIL levels and the HER2-EveNT score (Supplementary Fig. 21, Supplementary data 17, 18). In both trials, BCR and TCR read counts were highly correlated with the B cell lineage scores (rho = 0.95–0.97) and the T cell subtypes (rho ≥ 0.8), respectively. Instead, lower correlations between MCP-counter results and TIL levels were noted, with the highest value belonging to the B cell lineage in both studies (rho = 0.54 and 0.56 in NeoALTTO and CALGB 40601, respectively). These results are reassuring in that BCR/TCR measures highly correlate with B/T cells measured by MCP-counter, which may be expected as both are derived from RNA sequencing data, and suggest that MCP-counter fits better with BCR/TCR normalized reads than with TILs. BCR evenness was negatively correlated to the B cell lineage score, monocytic cells, as well as cytotoxic T cells (rho = −0.2 to −0.52), while a weak positive correlation (rho = 0.2 to 0.35) was found with neutrophils and endothelial cells in both studies. Therefore, we can hypothesize a relationship between the network of different cell types in the tumor microenvironment and B cell clonal expansion. In line with their suggested immunosuppressive role[38–40], neutrophils, endothelial cells and fibroblasts were positively correlated with the HER2-EvenNT score, while negative correlations with TILs were noted for endothelial cells and fibroblasts.

We next compared cell subtypes according to both pCR (ypT0/is ypN0) status and prognostic group (Supplementary Fig. 22, Supplementary data 19, 20) in NeoALTTO (N = 224 patients with prognostic score and breast + axilla pCR information available) and CALGB 40601 (N = 230). Patients in the same prognostic group and either pCR or RD showed similar profiles. In particular, remarkable differences were noted in both studies within the RD subgroup, where levels of different immune cells were significantly higher in the good prognosis group, highlighting the importance of immune variables in patients with RD.

## Gene expression profiling highlights biological differences in the prognostic groups

We next aimed to better characterize the gene expression features associated with the HER2-EveNT score by evaluating correlations with known signatures and performing comparisons between the prognostic groups. This in-depth characterization allowed us to identify the most relevant features captured by the score, including mostly processes associated to immune activation and suppression.

First, we evaluated the correlation between the HER2-EveNT score as a continuous variable and hallmark biological processes[41] scores computed with the gene set variation analysis (GSVA) tool[42] in the two studies (Fig. 5, Supplementary data 21, 23). Immune-related pathways, PI3K/AKT/mTOR signaling and proliferation (e.g., MYC targets, E2F targets) showed the highest negative correlation with the HER2-EveNT score. These results potentially describes the association of proliferation and HER2 downstream signaling with pCR (enriched in the good prognosis group with lower HER2-EveNT score), as previously suggested in NeoALTTO[10]. In addition, Notch and TGF beta signaling, as well as myogenesis, EMT and angiogenesis, which have been previously linked to immune evasion mechanisms and tumor progression[39,43–46], were positively correlated with the risk score and, thus, worse prognosis.

Furthermore, we performed differential gene expression analysis comparing the good and the poor prognostic groups in NeoALTTO and CALGB 40601 (Supplementary data 24, 25). Gene set enrichment analysis for hallmark biological processes performed on ranked genes from differential expression analysis showed comparable results to the GSVA analysis (Supplementary Fig. 23, Supplementary data 26). A gene ontology (GO) analysis on genes presenting a $\log_2$(fold change) > 0.58 or < −0.58 also revealed similar results, with immune-related GO processes associated with good prognosis (Supplementary data 27, 28). The top 30 biological processes associated with poor prognosis were more heterogeneous (Supplementary data 29, 30) and included, among others, neural biological processes (e.g., neuropeptide signaling pathway, nervous system process, nerve growth factor signaling pathway). Interestingly, neural processes have been recently found to be associated with lack of pCR and higher RD burden[47].

Among the differentially expressed genes with a $\log_2$(fold change) > 0.58 or < −0.58, 408 genes associated with good prognosis were in common in the two studies (Supplementary data 31). Shared immune-related genes with potential therapeutic relevance included CD274 [encoding programmed death-ligand 1 (PD-L1)], PDCD1 [encoding programmed cell death protein 1 (PD1)], CTLA4, CD38 (encoding an ectoenzyme involved in the extracellular adenosine production), IDO1 and ICOS, several genes coding for HLA class II proteins, and chemokines (i.e., CXCL9, CXCL10, associated to proinflammatory macrophages[18] and response to immunotherapy[48]).

Overall, these analyses show an immune-enriched phenotype for the good prognosis group and an enrichment in biological processes associated with a more aggressive phenotype and immune evasion in the poor prognosis group.

We next evaluated the relationship between the HER2-EvenNT score and known gene expression signatures. A pool of 709 gene expression signatures derived from the literature and single genes was computed on merged gene expression data after correction for study effect (Supplementary data 32, 33), allowing a more direct comparison between the two studies.

As shown in Supplementary Fig. 24, signatures negatively correlated with the risk score were mainly immune-related (Supplementary Data 34, 35). Of interest, among the signatures correlated (rho > 0.2) with a higher risk score, several were shared in the two studies, including signatures related to stemness, a lobular "reactive-like"

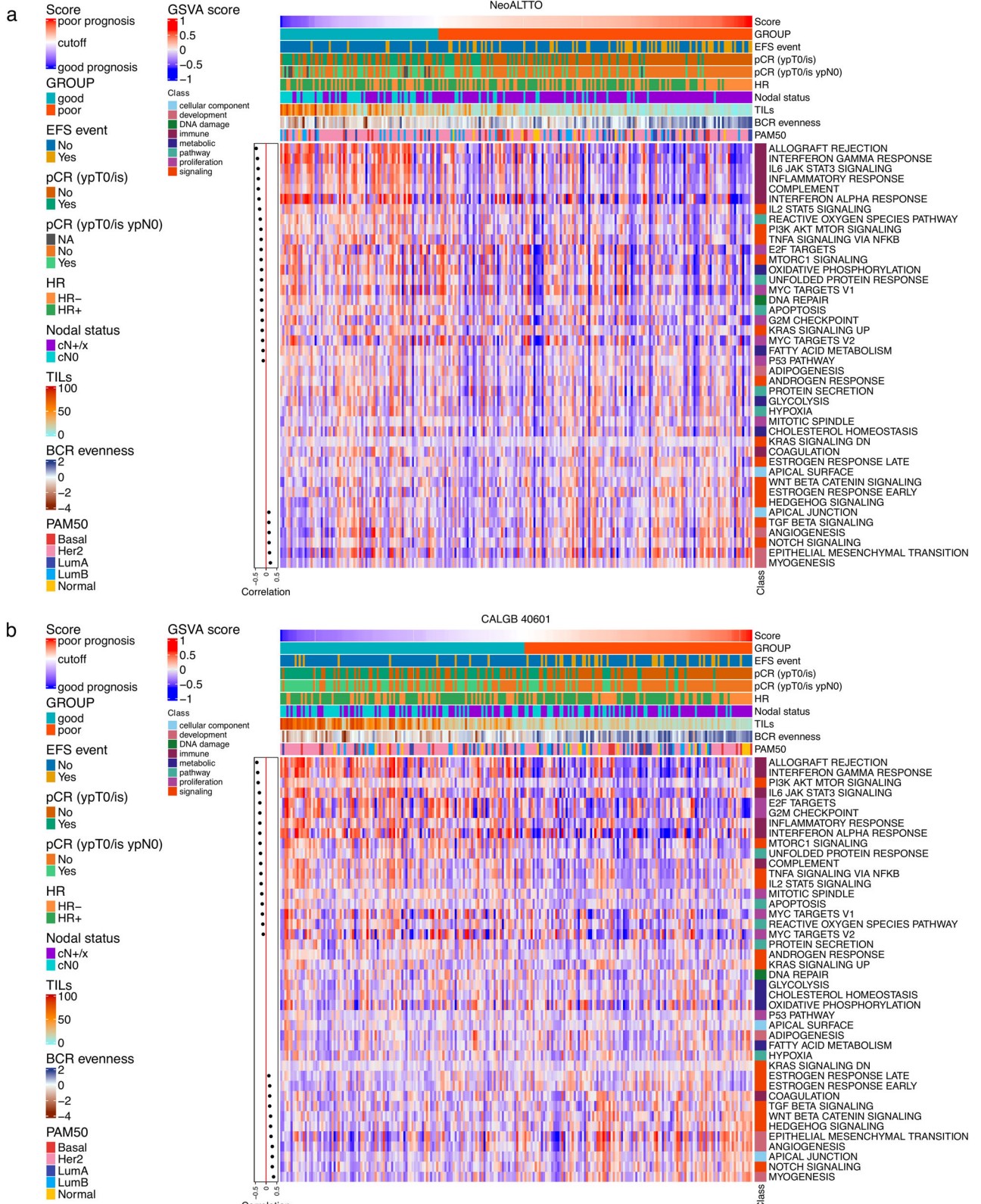

**Fig. 5 | Heatmap of GSVA scores for hallmark gene sets and correlations with the HER2-EveNT score in the NeoALTTO and CALGB 40601 datasets. a** Heatmap showing the GSVA scores for 42 hallmark gene sets in the NeoALTTO HER2-EveNT cohort ($N$ = 233). **b** Heatmap showing the GSVA scores for 42 hallmark gene sets in the CALGB 40601 HER2-EveNT cohort ($N$ = 230). Annotations on the top section of the heatmap include the HER2-EveNT score, the prognostic groups, presence or absence of EFS events, pCR, hormone receptor and clinical nodal status, TIL levels (%), BCR evenness (values as used in the model), and PAM50 subtypes. Cutoff

represents the value to divide the two prognostic groups (−1.3763). On the left side, Spearman correlations values between the GSVA scores and the prognostic score are shown if $P$ < 0.05 (two-sided). The red line divides positive and negative correlations. See also Supplementary data 23 and Source data. Basal basal-like, BCR B cell receptor, EFS event-free survival, GSVA gene set variation analysis, Her2 HER2-Enriched, HR hormone receptor, LumA luminal A, LumB luminal B, Normal normal-like, pCR pathological complete response.

signature[49], as well as a signature previously associated to lack of response to immune-checkpoint inhibitors[19].

Finally, we compared signature levels in patients categorized according to prognosis as defined by HER2-EveNT and pCR (ypT0/is ypN0) status (Supplementary Fig. 25, Supplementary data 36, 37), selecting 98 representative signatures/single genes describing immune processes, proliferation, as well as HER2 and estrogen/progesterone signaling. Both pCR and RD groups with good prognosis presented a clear enrichment in immune features. The poor prognosis samples, particularly those with RD, presented a lower immune-related signal instead. In addition, tumors achieving pCR, regardless of the prognosis, showed higher *ERBB2* and lower *ESR1* expression levels, and *PIK3CA* pathway activation signatures were more elevated in tumors with pCR and a good prognosis.

These findings suggest that intrinsic biological properties of the tumor (such as *ERBB2* and *ESR1* expression, in line with previous findings[5,6,10,50,51]) seem to be the primary driver of response to neoadjuvant treatments as measured by pCR in tumors belonging to the same prognostic group according to our model. In contrast, immune-related features are predominantly associated with prognosis, regardless of pCR status.

## Discussion

HER2-positive breast cancer represents a heterogeneous entity, as shown by PAM50 molecular subtypes[5,10]. From a therapeutic point of view, the interplay between the immune response and the activity of anti-HER2 agents is of crucial importance, as highlighted by evidence pointing to ADCC as a primary mechanism for the effect of trastuzumab[8]. Indeed, several studies have demonstrated an association between immune response (as described by TIL levels and/or gene expression signatures), pCR and prognosis in HER2-positive breast cancer[5,6,9–12,14,52–54]. However, the determinants of relapse after either pCR or RD have not been fully understood yet. In the present retrospective study, we adopted an approach aimed at describing both the "quantity" and "quality" of immune response in early-stage HER2-positive breast cancer by exploring the characteristics of baseline BCR and TCR repertoires in the NeoALTTO and CALGB 40601 phase III neoadjuvant clinical trials. We showed heterogeneity in BCR and, to a lesser extent, TCR measures according to hormone receptor status and PAM50 subtypes. Furthermore, we demonstrated the association between BCR repertoire measures describing clonal expansion suggestive of an antigen-specific response, namely lower evenness and higher Gini index, and better EFS in both NeoALTTO and CALGB 40601. We then built a model aimed at predicting EFS integrating clinicopathological characteristics, treatment response and immune-related features (i.e., TILs and BCR evenness). Of utmost importance, our model was developed in the NeoALTTO trial and independently validated in the CALGB 40601 study. By deriving the prognostic HER2-EveNT score from the model, we identified two groups of patients with distinct prognosis after neoadjuvant therapy, demonstrating an association with long-term EFS in the overall population as well as in the RD subgroup in both studies.

While pCR achievement is a strong prognostic factor in HER2-positive breast cancers after neoadjuvant therapy[27,28], the selection of further variables in the model suggests the importance of additional processes/features in determining prognosis. In fact, the risk of relapse is influenced by the interaction of several variables, particularly in the absence of pCR where the other biomarkers play a crucial role. The neoadjuvant setting presents some peculiar challenges in terms of biomarker discovery, as the relationship between prognostic features (related to survival/relapse risk) and predictive markers (related to pCR) may be complex. Indeed, some features such as proliferation, HER2 signaling or luminal phenotype may provide discordant information for pCR and survival (e.g., lower pCR rates and improved outcomes for luminal features)[6,55]. As additional

factors such as treatment administered could impact response to neoadjuvant therapies, pCR achievement may not be, still, fully captured by pre-treatment biomarkers, though progresses in this regard are being made[55].

Other predictive/prognostic models have been developed in early-stage HER2-positive breast cancer. For example, a combined score of TILs and tumor cellularity (CelTIL) measured at week 2 after starting neoadjuvant anti-HER2 therapy (without chemotherapy) provided predictive information in the PAMELA study[56] and was recently evaluated for long-term outcome prediction in the NeoALTTO trial[57]. Furthermore, a model including clinicopathological variables, TIL levels, PAM50 subtypes and genes was used to calculate a prognostic score (HER2DX) in the adjuvant Short-HER trial and evaluated in four neoadjuvant studies, including the CHERLOB and PAMELA trials, showing prognostic value[58]. More recently, this test has been refined and adapted to predict also pCR after pre-operative anti-HER2-based chemotherapy[55]. Importantly, our score was specifically designed and validated to predict prognosis in the neoadjuvant setting.

By characterizing the gene expression profiles of the prognostic groups, we showed an enrichment for immune-related processes in the group presenting good prognosis after either pCR or RD, leading us to hypothesize that tumors presenting an active immune response at baseline present biological similarities beyond HER2/estrogen receptor pathways activation.

As we are moving toward optimizing treatment strategies, we may argue that the lack of pCR in patients with baseline favorable clinical/immune features may not necessarily require treatment escalation approaches in all patients. In the seminal trial KATHERINE, post-operative trastuzumab emtansine (T-DM1) showed remarkable benefit compared to trastuzumab in patients with RD at surgery; however, more adverse events were also observed[59]. Identifying a population in which treatment escalation with T-DM1 is not beneficial would be very valuable both from a toxicity and a cost-effectiveness standpoint. In a biomarker analysis of KATHERINE, the benefit of T-DM1 appeared to be independent of the biomarkers assessed, including immune-related gene expression signatures and *ERBB2* expression levels computed essentially from RD samples[60]. Given evidence of transient gene expression changes induced by anti-HER2 drugs, it is not clear whether the same would be true of analyses performed on samples prior to drug exposure[61]. Based on our data, a combination of biomarkers performed on pre-treatment specimens may help identify a subset of patients (which according to our model may account for 20–30% of the patients with RD) with excellent outcomes and for whom post-operative trastuzumab alone could represent a viable option. Importantly, these patients can be identified computing the score with only the baseline features, as not achieving pCR will not have an impact on the prognostic category based on the HER2-EveNT score.

The role of B cell infiltration and clonal diversity in modulating the immune response in cancer has been described by several works[16–25]. The contribution of B cells to the response to immunotherapy is of interest in HER2-positive breast cancer, as several trials are currently testing immunotherapeutic strategies in this subtype[62]. Results in the advanced disease setting suggested a potential benefit for immunotherapy in pre-treated patients with PD-L1 positive tumors[63,64]. However, in the neoadjuvant setting, the primary analysis of the IMpassion050 trial did not show improvement in pCR rates when adding atezolizumab to standard treatment, and adverse events were more frequent in the atezolizumab arm[65]. These results highlight the necessity of implementing new immune biomarkers, and lead us to hypothesize that HER2-positive tumors with lower BCR evenness/higher Gini index and higher TIL levels (which can be related to higher levels of genes coding for immune checkpoints) could benefit from immunotherapy, considering that a precise selection of patients is crucial due to the potential associated toxicities. In addition, considering the importance of immune-related features and signatures in

patients with good outcomes regardless of pCR status, we may speculate that a potential benefit from immunotherapy in the neoadjuvant setting would translate mainly in longer EFS rather than increased pCR rates. Of note, the treatment arsenal of HER2-positive breast cancer is rapidly evolving, with several agents (e.g., antibody–drug conjugates and novel tyrosine kinase inhibitors) which are being evaluated in the early-stage setting[66] and may reshape the current biomarker landscape.

In light of all these aspects, our model could be prospectively tested in studies designed to evaluate approaches involving the use of immunotherapy-based regimens and/or de-escalation/shorter duration of post-operative treatments in low risk patients with an excellent prognosis, as well as alternative therapeutic strategies in those with poor outcome.

Our study presented several challenges and limitations. In addition to the already-discussed differences in terms of patient populations, the NeoALTTO and the CALGB 40601 trials had some differences in study design, namely in the timing of the start of neoadjuvant paclitaxel and in the post-operative treatments (details in the METHODS). We standardized the survival outcome measure (EFS) used for our analyses across the two trials; however, neither NeoALTTO nor CALGB 40601 were powered to detect survival benefits. Moreover, these trials evaluated lapatinib, which is not currently adopted in early-stage HER2-positive breast cancer. Therefore, validating these findings in cohorts receiving standard treatment regimens (e.g., pertuzumab in combination with trastuzumab) would be extremely valuable.

Characterizing the BCR and TCR repertoires from bulk RNA sequencing data represents a challenge in itself[15,67]. In line with previous reports[17,68], the number of reads mapping to BCR was higher than TCR, suggesting higher transcription of immunoglobulin mRNA compared to TCR mRNA, as well as a lower sensitivity in retrieving full TCR CDR3 sequences. Importantly, bulk RNA sequencing analysis does not allow to pair BCR/TCR chains (i.e., to identify the combination of heavy/light chains or alpha/beta chains forming a single BCR or TCR), contrary to single-cell sequencing technologies which enable to a better BCR/TCR as well as B and T cell characterization. In addition, RNA sequencing-based methods to describe BCR/TCR may provide a partial picture of the repertoires compared to more sensitive technologies for BCR/TCR sequencing[69]. These aspects may contribute to explain the moderate correlation of BCR/TCR read counts with TILs, although similar correlations have also been reported for other RNA sequencing-derived data such as immune-related gene expression signatures[12] and, as we showed, MCP-counter results. Differences in RNA sequencing pipelines between the two studies were also present, and we acknowledge that standardization of the methodology would be necessary to strengthen our findings further.

In addition, more reliable cutoffs may need to be defined for the HER2-EveNT score. Moreover, the inclusion of the residual cancer burden, given its prognostic value in patients with RD[70,71], as well as the integration of RD biological features, which may add an essential layer of information to the baseline characteristics, could further improve our model.

Despite these considerations, findings from NeoALTTO were confirmed in CALGB 40601, suggesting the robustness of the results, which first and foremost provide the rationale for interrogating more extensively the role of B cells in this disease.

In conclusion, we demonstrated the heterogeneity of BCR repertoire measures and immune response within HER2-positive breast cancer and the potential of BCR diversity as a prognostic biomarker. Our model identified a group of patients with immune-enriched tumors who may be eligible for treatment de-escalation approaches, even after RD, and highlighted the importance of integrating clinicopathological characteristics, treatment response information, and immune-related features to define the clinical risk in the neoadjuvant

setting. Further validation of our findings and exploration of the role of BCR/TCR repertoire measures in HER2-positive as well as other breast cancer subtypes is warranted.

## Methods

### NeoALTTO and CALGB 40601 study designs and patient populations

The NeoALTTO phase III trial randomized 455 HER2-positive early-stage BC patients to receive neoadjuvant trastuzumab (T), lapatinib (L) or T + L[1,3,4]. After 6 weeks of anti-HER2 treatment, weekly paclitaxel was added for further 12 weeks before surgery. Lapatinib dose was reduced during the paclitaxel administration after a protocol amendment due to toxicity. In the adjuvant phase, anthracycline-based chemotherapy (fluorouracil, epirubicin and cyclophosphamide) for three cycles was administered, followed by the same anti-HER2 therapy received in the neoadjuvant phase for a total duration of 52 weeks. Eligible patients had HER2-positive primary breast cancer with a minimum tumor size of 2 cm and adequate cardiac function. Patients were recruited between January 2008 and May 2010.

The primary endpoint of the trial was pCR defined as either absence of invasive tumor cells in the breast (ypT0/is) as defined by the National Surgical Adjuvant Breast and Bowel Project criteria, later amended to the absence of invasive tumor cells in the breast and in the axillary lymph nodes (ypT0/is ypN0) according to the Food and Drug Administration criteria. The main secondary endpoint was EFS, defined as the time from randomization to first event, i.e., breast cancer relapse after surgery, second primary malignant neoplasm, or death without recurrence for women who received surgery for breast cancer, or, for those who did not undergo surgery, death during clinical follow-up or non-completion of any neoadjuvant investigational drugs due to progressive disease. OS was defined as the time from randomization to death from any cause.

In the phase III CALGB 40601 trial, 305 eligible patients with newly diagnosed, histologically confirmed, untreated clinical stage II to III HER2-positive disease were randomized to receive neoadjuvant weekly paclitaxel in combination with either T, L, or T + L for 16 weeks[5,6]. After surgery, treatment with dose-dense doxorubicin and cyclophosphamide, as well as completing 1 year of trastuzumab were recommended for all patients. Patients were recruited between December 2008 and February 2012. Based on non-inferiority reports and higher toxicity, the L arm was closed early. Pathological complete response (defined as ypT0/is) was the study's primary endpoint, while secondary endpoints included relapse-free survival and OS. To make survival outcomes comparable between the two studies, EFS was also assessed in the CALGB 40601 using the same definition adopted in the NeoALTTO study. Patient characteristics, designs and results of the two trials have been previously published[1,3–6]. The cohorts with RNA sequencing data analyzed in the present work have also been previously described[5,6,10].

Ethics committee and relevant health authorities at each participating site approved the NeoALTTO and CALGB 40601 studies and all patients provided written informed consent including future biomarker research. For NeoALTTO, the sites which run the trial are listed in Supplementary data 38. For the CALGB 40601 trial, the protocol was central IRB approved, being the University of North Carolina the lead Institution and Dr. Lisa Carey the study chair. The current combined analysis has been approved by the TransALTTO scientific committee and Alliance Publications Committee, and was conducted in accordance with the Declaration of Helsinki.

The NeoALTTO trial is registered at www.clinicaltrials.gov as NCT00553358, while the CALGB 40601 trial is registered as NCT00770809.

In NeoALTTO, clinical data were collected at Institut Jules Bordet and Frontier Science Scotland; analyses are based on the on the clinical study database frozen on May 26, 2016. In CALGB 40601, clinical data

were collected at the CALGB (Alliance) Statistics and Data Center; analyses are based on the on the clinical study database frozen on October 06, 2021.

## Samples collection and processing, TILs evaluation

Out of the 455 patients enrolled in the NeoALTTO trial, baseline frozen tumor biopsies were available for 423 patients. RNA was successfully sequenced from frozen tumor samples in 254 patients as previously described[10].

In the CALGB 40601 trial, RNA sequencing data from baseline frozen tumor samples were available for 264 patients, as previously reported[5,6].

Stromal TILs were scored as % following the International TILs Working Group guidelines[36] in both studies, with % referring to the area occupied by mononuclear inflammatory cells over the total intratumoral stromal area. Results for TIL levels in the NeoALTTO trials have been previously published[9]. In CALGB 40601, TILs were scored by a pathologist (B.S.), and the average of the TIL scores was considered when more than 1 slide/sample was available, as previously described[12].

## RNA sequencing data processing

RNA sequencing processing for baseline frozen biopsy samples in the two trials has been previously described[6,10]. RNA sequencing was performed at the BRIGHTcore sequencing facility of the Université Libre de Bruxelles for NeoALTTO, and at the University of North Carolina High-Throughput Sequencing Facility for CALGB 40601. For BCR/TCR analyses, immune-deconvolution, differential gene expression, and gene set enrichment analyses (i.e., all analysis specifically conducted in each study separately) starting from BAM files obtained in the NeoALTTO and CALGB 40601 studies, read pairs were trimmed using Trimmomatic v0.39[72], and Salmon v1.5.1[73] was used for alignment to the human reference GRCh38/hg38 (patch 13), using GENCODE v38 for the gene positions. Reads in NeoALTTO were 100-bp long, while in CALGB 40601 reads were 50-bp long. There were more reads mapping to genes in CALGB 40601 (51.8 million reads/sample on average) than in NeoALTTO (15.4 million on average).

When evaluating the potential technical differences between NeoALTTO and CALGB 40601 in terms of number of normalized BCR and TCR reads and clones, we considered the difference in read length as potential cause of such differences, and explored whether and how BCR/TCR read/clone counts would be impacted in NeoALTTO by trimming reads from 100-bp to 50-bp (Supplementary data 1). Since the RNA sequencing was paired-end, the in-silico trimming from 100-bp to 50-bp was done from the middle of the fragment, representing what would be obtained by sequencing at 50-bp.

The "tximport" R package (v1.16.1)[74] was used to obtain gene-level estimates and transcript per million (TPM) normalized gene expression levels.

Intrinsic subtypes were obtained from RNAseq gene expression data in both studies by applying a HER2/estrogen receptor subgroup-specific gene normalization method[6,75] followed by the PAM50 predictor[13]. For this objective, aiming to ensure uniformity in the PAM50 analyses we followed the workflow described in Fernandez-Martinez et al.[6] and performed a study effect corrected analysis of the RNA sequencing data. In detail, purity-filtered reads were aligned to the human reference GRCh38/hg38 genome using STAR v2.4.2a[76]. Transcript (GENCODE v22) abundance estimates were generated by Salmon v0.6.0[73] in 'quant' mode, based on the STAR alignments. Raw read counts for all the RNA sequencing samples were normalized to a fixed upper quartile[77]. Normalized gene counts were then $\log_2$ transformed (zeros were unchanged), and genes were filtered for those expressed in 70% of samples in both studies. The gene expression data was then median-centered by study and ultimately, all the gene expression values were standardized.

## BCR and TCR repertoire diversity measures

BCR/TCR repertoires were extracted from bulk RNA sequencing data using the MiXCR tool[29,30] (v3.0.13) with default parameters, specifying the option "OallowPartialAlignments = true", and including all immunoglobulin and TCR chains. Briefly, we aligned sequencing reads to reference V, D, J and C genes of T- or B- cell receptors. Clonotypes were assembled using aligned reads to extract the CDR3, which is the region conferring the antigen recognition capability[15]. The option "OallowPartialAlignments" allowed to rescue incomplete alignments related to the absence of V or J CDR3 part, as the tool builds contigs from different initial alignments while avoiding artificial diversity generation[30]. Indeed, this enables to assemble alignments only partially covering the CDR3 region, which is a possible occurrence due to the random coverage of BCR/TCR segments by fragmented reads in RNA sequencing. In particular, clones were defined as having the same CDR3 nucleotide sequence. Moreover, we tried to assign each clone to a specific type of BCR (kappa, lambda or heavy) or TCR (alpha, beta, gamma, delta) chain, based on the V, D, and J genes. For some clones MiXCR gives multiple matches for some or all genes. If all genes pointed to the same chain type, the clone was assigned to that type. In the case of a clone with a gene with discordant chain types (e.g., V genes *TRAV17* and *TRBV5*), the other gene of the clone was checked, and its chain type was used if it was univocal and compatible with the other gene (e.g., *TRAJ50*). In the cases where discordant chains were found between the genes (e.g., *TRAV24* for V gene and *TRBJ2-5* for J gene) or where it was impossible to determine univocally the chain type, the clone was kept and considered as either BCR or TCR, but its chain type was left as missing.

Isotypes (IgG, IgA, IgM, IgD, IgE) for the BCR heavy chain were determined based on the constant region (C region) information, whenever present. In particular, the isotype could not be determined in 23% of the IGH clones in NeoALTTO and 19% in CALGB 40601. For the remaining IGH clones, isotype proportions are shown in Supplementary Fig 3e–f.

The following BCR and TCR repertoire metrics were calculated: read counts, number of clones, length of the CDR3, top and second top clone proportions. Diversity indices including Gini index, Gini-Simpson index and species evenness were calculated as well. Those measures were computed for each BCR/TCR chain separately, as well as "global" measures considering all clones belonging to either BCR or TCR (Supplementary data 2, 3).

Evenness and second top clone were calculated if at least two different clones were present. Gini and Gini-Simpson indices were calculated when at least 1 clone was present, assigning a value of 0 when only one clone was present. CDR3 length was calculated when at least one clone was present. We defined BCR/TCR metrics (illustrative examples in Supplementary Fig. 1) as follow: read count reflects the number of reads mapping to CDR3 regions of BCR or TCR, and was normalized (i.e., divided) by the total number of reads mapping to the transcriptome in each sample and multiplied by a factor of 1000; the number of clones is defined as the total number of different clones in the sample; the CDR3 length is calculated as mean of the number of nucleotides included in that region; top and second top clone proportions measure the proportion of reads mapping to the first and second most expressed clone in the BCR/TCR population for that sample; Gini index describes the degree of inequality among values within the BCR/TCR population and was calculated using the function "gini" from the "reldist" R package (v1.6-6)[78]; Gini-Simpson index is determined by the total number of clonotypes and their relative abundance, and describes the probability that two entities are from different types; evenness is Shannon entropy normalized by the number of clones and describes the uniformity of the clonotypes, measuring distribution equality. Gini-Simpson index was calculated as $1 - \sum p_i^2$, while evenness was calculated as $-\sum p_i \log_{Nclones}(p_i)$, with $p_i$ being the fraction of each clone.

From the MiXCR output, we calculated these measures for each chain forming BCR and TCR (IGH/K/L and TRA/B/D/G, respectively), as well as global values from the total reads mapping to BCR and TCR, thus representing the whole BCR and TCR repertoires.

For univariable and multivariable analyses, the Shapiro-Francia test was used to evaluate normality of the BCR and TCR measures separately in NeoALTTO and CALGB 40601. In details, when a BCR/TCR feature had no values equal to 0 in the cohort, a log transformation ($\log_{10}$) was applied, while if 0 s were present we log transformed the value after adding a small constant defined as the 10% quantile of the value divided by 10. The distribution of the newly obtained values was compared to the original distribution, and if a $P$ value > 0.05 (Shapiro-Francia test) was obtained, the variable was not log transformed as normality was not improved. Furthermore, for this analysis, centering and scaling were applied independently in the two studies to the BCR and TCR measures, first removing their mean and then dividing by their standard deviation.

## Development of the prognostic model

A forward stepwise approach was used to build the prognostic model. In particular, starting from a null Cox model for EFS, variables selected from a pool were added following the AIC method [function "steps" from the "stats" R package (v4.0.5)[79]]. BCR and TCR measures were included in the variable selection process, together with treatment response [pCR status, including both breast pCR (ypT0/is) and breast + axilla pCR (ypT0/is ypN0)], hormone receptor and estrogen receptor status, stromal TIL levels (as continuous variable), PAM50 subtypes (HER2-E vs. others), tumor size (cT2 vs. ≥ cT3), clinical nodal status (cN0 vs. cN+/x), treatment arm (single arms vs. combination), and age (as continuous variable). A set of gene expression signatures which showed predictive potential in NeoALTTO, particularly in the combination arm[10] (2 immune signatures[80,81], 2 stroma signatures[80,82], and 2 proliferation signatures−Genomic Grade Index[83] and AURKA[80]; genes included and coefficients are available in Supplementary data 39), an IgG signature previously associated with improved outcome in CALGB 40601[6,84], as well as expression levels of *ERBB2* and *ESR1* genes were also included in the variables tested. The list of variables, steps and corresponding AIC values, as well as the details of the final model are shown in Supplementary data 12.

Before calculating the signature scores, genes with a total read count across all samples < 20 were excluded, and normalization was performed by dividing gene counts by the sample mean gene count followed by $\log_{10}$ normalization after adding a small constant (0.001). Then, signature scores were computed as the weighted mean of the log-expressions of their genes, with gene-specific weights equal to +1 or −1 based on the direction of their association with the phenotype in the original publication or to specific coefficients for the AURKA signature.

With regards to BCR and TCR measures, the same procedure followed for uni- and multi-variate analysis was applied in NeoALTTO (i.e., transformation according to the Shapiro-Francia test, then centering and scaling).

The model was developed on a cohort of 221 patients from the NeoALTTO trial with all data available. Subsequent analysis regarding the prognostic score in NeoALTTO were then performed in a cohort of 233 patients with all the variables selected for the model available, except for the analyses including pCR in breast + axilla (ypT0/is ypN0), which were performed on 224 patients.

A prognostic score (HER2-EveNT) was computed multiplying the coefficient obtained for each variable in the final Cox model (Supplementary data 12) for the value of the same variable, and summing up the obtained values. For categorical variables, a value of 0 or 1 was used according to their dichotomized status (i.e., pCR = 1, RD = 0; HR+ = 1, HR− = 0; cN0 = 1; cN+/x = 0).

The score was therefore calculated as:

BCR evenness (standardized value) × 0.300436587 + TILs (%) × −0.024107504 + pCR (0 or 1) × −0.568111234 + HR status (0 or 1) × −0.525011259 + nodal status (0 or 1) × −0.504914127.

As the coefficients are the natural logarithm of the HRs for each variable, the score represents a measure of the relative risk of relapse calculated based on the values of each variable in the single patient, with a lower score being associated with a better prognosis. Variables with a negative coefficient were associated with improved EFS, while a positive coefficient was associated with worse prognosis.

This score was then divided into tertiles. As EFS > 90% was reached for the first tertile in NeoALTTO, the cutoff between the first and the intermediate tertiles was used to define the two final prognostic groups. An external independent validation was performed testing the model in 230 patients from the CALGB 40601 trial with pCR, hormone receptor, clinical nodal status, TILs, and BCR evenness data available. As BCR evenness was not log transformed in NeoALTTO, the BCR evenness values in CALGB 40601 were only centered by removing the mean of NeoALTTO BCR evenness (0.7917893) and scaled dividing the obtained values by the NeoALTTO BCR evenness standard deviation (0.1004482) before computing the HER2-EveNT score.

The concordance index (Harrel's C-index) was assessed to estimate the predictive performance of the model. Cutoffs derived from NeoALTTO were applied on the score calculated in CALGB 40601 to identify correspondent prognostic groups.

## Immune-deconvolution, gene set enrichment analysis, differential expression analysis, gene ontology analysis

The MCP-counter tool (v1.2.0)[37] was applied on TPM-normalized RNA sequencing data using the "immunedeconv" R package (v2.0.3)[85]. Abundance scores of eight immune populations (CD3 + T cells, CD8 + T cells, cytotoxic lymphocytes, natural killer cells, B cell lineage, cells originating from monocytes, myeloid dendritic cells, and neutrophils), as well as two stromal cell types (endothelial cells and fibroblasts) were estimated.

In order to evaluate biological pathways at the single-sample level and their correlation with the prognostic score as a continuous variable, we applied GSVA (v1.36.3)[42] (with "method = GSVA") to the variance stabilized expression value matrices [obtained with the function "vst" in the "DEseq2" R package (v1.28.1)[86], with option blind = FALSE] in each study separately (patients with HER2-EveNT score available, $N = 233$ in NeoALTTO and $N = 230$ in CALGB 40601), to compute sample-wise gene set enrichment scores. The objective here was to evaluate within each cohort the correlation with the prognostic score, rather than to compare directly the two studies. We selected 42 hallmark gene sets[41] (version 7.4) downloaded from MSigDB (https://www.gsea-msigdb.org/gsea/msigdb/) with the "msigdbr" R package (v7.4.1)[87], excluding the following eight hallmark signatures due to the lack of association with tumor processes or microenvironment: peroxisome, pancreas beta cells, spermatogenesis, bile acid metabolism, heme metabolism, UV response up, UV response down, xenobiotic metabolism. We computed the correlation of the GSVA scores with the prognostic score as continuous value (considering as significant correlations with $P < 0.05$). In Fig. 5, classes of hallmarks are defined according to the original publication[41].

The differential gene expression analysis between good and poor prognosis groups was performed using DEseq2 on gene-level abundance estimates from Salmon (functions "DESeqDataSetFromTximport" and "DESeq"). These analyses were performed on the NeoALTTO and CALGB 40601 data separately. Genes with low expression levels (sum of the reads across all samples < 10) were removed before the analysis. In order to identify genes and pathways/biological processes associated to prognosis according to our model, prognostic groups in the whole population ($N = 233$ in NeoALTTO and $N = 230$ in CALGB 40601) were compared. Genes

were considered up- or down-regulated in the good prognosis group if the fold change was either positive or negative, respectively, specifying "alpha = 0.05" in the function "results". The "lfcShrink" function was used to perform $log_2$(fold change) shrinkage according to the "apeglm" method[88]. Genes were considered significantly differentially expressed for FDR < 0.05.

Gene set enrichment analysis comparing the prognostic groups was performed applying the "fgsea" R package (v1.14.0)[89] on the DEseq2 results, ranking genes by $sign[log_2(fold\ change)] \times -log_{10}(P\ value)$. The same 42 hallmark gene sets previously mentioned were evaluated. Gene sets with normalized enrichment score (NES) > 0 and FDR < 0.05 were considered upregulated in the good prognosis groups, while gene sets with NES < 0 and FDR < 0.05 were considered downregulated.

Gene ontology analysis on the "biological process" domain was performed on all the identified differentially expressed genes with FDR < 0.05 and $log_2$(fold change) of > 0.58 or < −0.58 (i.e., fold change > 1.5 or < 0.67) using the "topGO" R package (v2.40.0) (with nodeSize = 5)[90].

### Gene expression signature analysis on integrated RNA sequencing data

A collection of 709 gene expression signatures derived from the literature and partially summarized before[6,84,91] was computed on merged gene expression data after batch effect correction by study, obtained as previously described in the METHODS section when discussing PAM50 intrinsic subtype harmonization. This analysis allowed a more direct comparisons between the prognostic groups in NeoALTTO and CALGB 40601, ensuring the robustness of the findings described when evaluating each study separately.

The signature scores were computed by calculating the median expression of all the genes within a signature. Multiple immune-related biomarkers were included in this gene-signature collection, most of them initially extracted by comparing the gene expression pattern of different immune cell sub-populations[92,93]. Other immune signatures were obtained by an unsupervised cluster of different breast cancer samples as previously described[26,84]. The list of gene expression signatures with the PMID of the reference and the genes included in each signature is available in Supplementary data 40.

Signature scores were then centered removing their mean and scaled dividing them by their standard deviation in the two studies. We next computed the correlation of the signature scores with the prognostic score as continuous value (considering as significant correlations with P < 0.05). We then selected 98 signature scores/single gene expression levels to compare four groups created based on the prognostic score and pCR status (ypT0/is ypN0), i.e., "pCR, good prognosis", "pCR, poor prognosis", "RD, good prognosis", "RD, poor prognosis". The effect size was obtained by applying a linear regression model and the P value derived from a Wilcoxon rank sum test, comparing each group against the rest.

### Statistical analysis

The Reporting Recommendations for Tumor Marker Prognostic Studies criteria were followed for this study[94].

Univariable and multivariable [controlling for tumor size (≥ cT3 vs. cT2 in NeoALTTO, ≥ cT3 vs. cT1-2 in CALGB 40601), nodal status (cN0 vs. cN+/x), hormone receptor status, age as a continuous variable, treatment arm (T or L vs. T + L), PAM50 subtypes (HER2-E vs. others)] Cox proportional hazard models were used for survival analysis. Logistic regressions were used to compute P values for pCR. In univariable analysis for survival, P values were obtained with the likelihood ratio test. For multivariable analyses, P values were derived by an ANOVA on nested logistic and Cox models. For each variable tested, univariable and multivariable analyses were performed including patients with available data. In CALGB 40601, patients with tumor size

not available (N = 16/264) were not included in the multivariable analysis controlling for clinicopathological characteristics. In forest plots, hazard ratios/odds ratios and confidence intervals for continuous variables were computed after centering the variable by removing its mean and scaling by dividing the variable by its standard deviation.

Wilcoxon rank sum (for comparisons between two groups) and Kruskal-Wallis tests (for comparisons between three or more groups) were used to compare continuous variables according to categorical variables. Fisher's test was performed to compare categorical variables. All correlations were assessed calculating the Spearman's rank correlation coefficient (rho) on pairwise complete observations, and considered significant for P < 0.05.

Kaplan–Meier survival curves were used to represent EFS according to prognostic groups, and P values obtained with log-rank test.

All P values were two-sided. False discovery rates (FDRs) were obtained adjusting P values with the Benjamini & Hochberg method, whenever specified. When multiple comparisons were performed (e.g., BCR/TCR according to hormone receptor status and PAM50, gene signatures or MCP-counter values in different groups), P values obtained with Wilcoxon rank sum and Kruskal-Wallis tests were adjusted on all related comparisons, in the two studies separately. P values and FDRs were considered significant when < 0.05.

Heatmaps were obtained with the "ComplexHeatmap" R package (v2.4.3)[95]. For the boxplots, the boxes are defined by the upper and lower quartile; the median is shown as a bold colored horizontal line; whiskers extend to the most extreme data point which is no more than 1.5 times the interquartile range from the box.

All statistical analyses were performed using the R software (v4.0.5)[79]. The analyses in the present manuscript were performed at the Institut Jules Bordet/Université Libre de Bruxelles and Lineberger Comprehensive Cancer Center/University of North Carolina.

### Reporting summary

Further information on research design is available in the Nature Portfolio Reporting Summary linked to this article.

### Data availability

For reproducibility purposes, the RNA sequencing data (fastq files) at baseline from NeoALTTO have been deposited to the European Genome-phenome Archive (EGA) database under accession number EGAS00001007563. The data can be obtained upon signature of a data access agreement between the investigator requesting the access and Institut Jules Bordet (IJB), subject to applicable laws. For reproducibility purposes, the NeoALTTO clinical data at IJB can be obtained upon signature of a data transfer agreement between the investigator and IJB, subject to applicable laws. The access to these data can be requested by contacting the corresponding Author (christos.sotiriou@hubruxelles.be). For investigators aiming to perform original research, the NeoALTTO RNA sequencing data at baseline and the clinical data are available upon request after submission of a research project proposal (RPP) to the RPP's administrator (alttoresearchproposals@frontier-science.co.uk). In detail, access to data for research will be granted upon review of the RPP and its endorsement by the study Steering Committee, and after entering into an appropriate data access agreement between BIG, IJB, and the investigator, subject to applicable laws. More details and documents required can be found at https://bigagainstbreastcancer.org/clinical-trials/neoaltto/ under the section "Translational Research". The policy for access to residual biological samples and data in the NeoALTTO study is a fair scientific review process set up to ensure precious biological samples or data collected in the studies are accessed appropriately, to avoid duplication of efforts and foster collaboration. The data from the study are not anonymized yet, only pseudonymized, therefore they are still considered identifiable, and cannot be made publicly available at this

point. In order to ensure that they are shared in a way that preserves the privacy of patients and complies with the relevant laws and regulations including the European General Data Protection Regulation (GDPR), researchers can only access the data after they sign the data transfer agreements mentioned above, either for reproducibility or for original research purposes. Gene expression data from CALGB 40601 is deposited in Gene Expression Omnibus (GEO) under accession code GSE116335. Fastq files (RNA sequencing) and clinical data, including TIL levels, from CALGB 40601 are available at the NCBI database of Genotypes and Phenotypes (dbGaP) repository under accession code phs001570.v3.p1 (https://www.ncbi.nlm.nih.gov/projects/gap/cgi-bin/study.cgi?study_id=phs001570.v3.p1). The data are available under controlled access to ensure that they are shared in a manner consistent with the research participants' informed consent, and that the confidentiality of the data and the privacy of participants is protected. Principal investigators seeking access to dbGaP datasets can request them through the controlled-access portal. More information can be found at https://dbgap.ncbi.nlm.nih.gov/aa/wga.cgi?page=login. Clinicopathological data from NeoALTTO and CALGB 40601 can be obtained as specified above. Other data supporting the findings of this study are available within the article, supplementary information files and supplementary data. MSigDB is available at https://www.gsea-msigdb.org/gsea/msigdb. Source data are provided with this paper.

## Code availability
The custom script used to generate to BCR/TCR measures from MiXCR output is available at https://github.com/BCTL-Bordet/BCR_TCR_analyses.

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

## Acknowledgements

NeoALTTO is a BIG and SOLTI led trial. CALGB is now part of the Alliance for Clinical Trials in Oncology. Related to the NeoALTTO trial, research reported in this publication was supported by the Fondation contre le cancer, the Breast Cancer Research Foundation, Association Jules Bordet, and the Belgian Fonds National de la Recherche Scientifique (F.R.S-FNRS). Related to the CALGB 40601 trial, research reported in this publication was supported by the National Cancer Institute of the National Institutes of Health under Award Numbers U10CA180821, U10CA180882, and U24CA196171 (to the Alliance for Clinical Trials in Oncology), UG1CA233180, UG1CA233337, UG1CA233373, R01-CA229409, Breast Cancer Research Foundation. The conduct of the NeoALTTO study was funded by GlaxoSmithKline and later Novartis. The RNA sequencing in NeoALTTO, on which part of the analyses described in this manuscript are based, was funded by GlaxoSmithKline. Both trials received support from GlaxoSmithKline, who provided the study drug. For part of the analysis, computational resources have been provided by the Consortium des Équipements de Calcul Intensif (CÉCI), funded by the Fonds de la Recherche Scientifique de Belgique (F.R.S.-FNRS) under Grant No. 2.5020.11 and by the Walloon Region. M.R. was supported by Télévie and the Belgian Fonds National de la Recherche Scientifique (F.R.S.-FNRS) and by Fondation Rose et Jean Hoguet. R.S. is supported by the Breast Cancer Research Foundation (BCRF, grant nr. 17-194). The content is solely the responsibility of the authors and does not necessarily represent the official views of the National Institutes of Health.

## Author contributions

M.R., D.V., F.R., C.S. conceived and designed the study. M.R., A.F.-M., D.V. contributed to develop the methodology. J.C., K.V.B., D.W.H. provided statistical support. B.S., E.P.W., S.E.-A., M.P., S.DC., W.F.S., I.E.K, R.S., S.L., L.P., C.M.P., L.A.C. contributed to data acquisition and to the design of the study. M.R., A.F.-M., D.V., F.R., C.M.P., L.A.C., C.S. analyzed and interpreted the data. M.R., A.F.-M., D.V., K.A.H., J.S.P. performed the analyses. D.V., K.A.H., J.S.P. provided bioinformatics support. C.M.P., L.A.C., F.R., C.S. supervised the analyses and the study. M.R., A.F.-M., D.V., L.A.C., C.S. prepared the manuscript, which was reviewed and approved by all authors.

## Competing interests

The authors declare the following competing interests. J.S.P.: equity and consulting from Reveal Genomics; royalties from patent from Veracyte. B.S.: participation in speaker's Bureau for AstraZeneca. E.P.W.: honoraria and equity, board member for Oncoclinicas; consultant honoraria from Carrick Therapeutics, GSK, Jounce Therapeutics; Consultant/Honoraria Research to Institute from Genentech/Roche; honoraria from Genomic Health; scientific advisory board honoraria from Leap Therapeutics. S.E.-A.: grant from Novartis within the submitted work and from Roche/Genentech and Pfizer outside the submitted work. M.P.: invited speaker for AstraZeneca, Lilly, MSD, Novartis, Pfizer, Roche-Genentech; consultant for Camel-IDS/Precirix, Roche-Genentech; advisory board for Frame Therapeutics, Gilead, Immunomedics, Immutep, Menarini, NBE Therapeutics, Odonate, Roche-Genentech, SeaGen, Seattle Genetics; member of boards of directors, scientific board for Oncolytics; research grants to her Institution from AstraZeneca, Immunomedics, Lilly; funding to her Institution from Menarini, MSD, Novartis, Pfizer, Radius, Roche-Genentech, Servier, Synthon. S.DC.: fees for medical education from Novartis, Pierre-Fabre, and IQVIA; institutional grant IG 20774 of Fondazione Associazione Italiana Ricerca contro il Cancro (AIRC); "ad hoc" medical advisor for Medica Scientia Innovation Research (MEDSIR), Barcelona (Spain). W.F.S.: founder stock in Delphi Diagnostics and publicly traded stock in IONIS Pharmaceuticals and Eiger BioPharmaceuticals; consultant/advisor to Merck, AstraZeneca; co-inventor of pending patent "Targeted Measure of Transcriptional Activity Related to Hormone Receptors", United States, Provisional Patent Application Serial No. 62/329,774; support for unrelated research from Pfizer;

uncompensated scientific advisor to Delphi Diagnostics. I.E.K.: advisory board participation/consultant and received honoraria from Bristol Meyers Squibb, Daiichi/Sankyo, Macrogenics, Genentech/Roche, Seagen, AstraZeneca, Novartis, Merck; institutional research funding/grants (paid to Institution) from Genentech/Roche, Pfizer, Macrogenics. S.L.: research funding to her institution from Novartis, Bristol Meyers Squibb, Merck, Puma Biotechnology, Eli Lilly, Nektar Therapeutics, AstraZeneca, Roche-Genentech and Seattle Genetics; consultant (not compensated) to Seattle Genetics, Novartis, Bristol Meyers Squibb, Merck, AstraZeneca, Eli Lilly, Pfizer, Gilead Therapeutics and Roche-Genentech; consultant (paid to her institution) to Aduro Biotech, Novartis, GlaxoSmithKline, Roche-Genentech, AstraZeneca, Silverback Therapeutics, G1 Therapeutics, PUMA Biotechnologies, Pfizer, Gilead Therapeutics, Seattle Genetics, Daiichi Sankyo, Merck, Amunix, Tallac Therapeutics, Eli Lilly and Bristol Meyers Squibb. L.P.: consulting fees and honoraria from Seagen, Pfizer, AstraZeneca, Merck, Novartis, Bristol-Myers Squibb, Pfizer, Genentech, Eisai, Pieris, Immunomedics, Clovis, Syndax, H3Bio, Radius Health, Personalis, Daiichi, Natera and institutional research funding from Seagen, AstraZeneca, Merck, Pfizer and Bristol Myers Squibb. R.S.: non-financial support from Merck and Bristol Myers Squibb (BMS); research support from Merck, Puma Biotechnology and Roche; personal fees from Roche, BMS and Exact Sciences for advisory boards. C.M.P.: equity stock holder and consultant of Bio-Classifier LLC; listed as an inventor on patent applications for the Breast PAM50 Subtyping assay; equity stock holder and consultant of Reveal Genomics. L.A.C.: research funding to her Institution from Sindax, Novartis, NanoString Technologies, Seattle Genetics, Veracyte, AstraZeneca; Royalty-sharing agreement, investorship interest in licensed intellectual property to startup company, Falcon Therapeutics, that is designing neural stem cell–based therapy for glioblastoma multiforme (immediate family member); uncompensated relationship for Eisai, Sanofi, Lilly, SeaGen, Novartis (Institution), G1 Therapeutics (Institution), Genentech/Roche (Institution), GlaxoSmithKline (Institution), AstraZeneca (Institution), Daiichi Sanyo (Institution), Exact Sciences (Institution). C.S.: advisory board (receipt of honoraria or consultations fees) for Astellas, Cepheid, Vertex, Seattle genetics, Puma, Amgen, Exact Sciences; participation in company sponsored speaker's bureau for Eisai, Prime Oncology, Teva, Foundation Medicine, Exact Sciences; other support (travel, accommodation expenses) from Roche, Genentech, Pfizer. The remaining authors declare no non-financial or financial competing interests.

## Additional information

[1]Breast Cancer Translational Research Laboratory, Institut Jules Bordet, Hôpital Universitaire de Bruxelles (H.U.B), Université Libre de Bruxelles (ULB), Brussels, Belgium. [2]Lineberger Comprehensive Cancer Center, University of North Carolina, Chapel Hill, NC, USA. [3]White Plains Hospital, White Plains, NY, USA. [4]Alliance Statistics and Data Management Center, Mayo Clinic, Rochester, MN, USA. [5]Alliance Statistics and Data Management Center, Weill Cornell Medicine, New York, NY, USA. [6]Yale Cancer Center, Yale School of Medicine, New Haven, CT, USA. [7]Breast International Group, Brussels, Belgium. [8]Medical Oncology Department, Institut Jules Bordet and l'Université Libre de Bruxelles (U.L.B.), Brussels, Belgium. [9]Integrated biology platform unit, Department of Applied Research and Technological Development, Fondazione IRCCS Istituto Nazionale dei Tumori, Milan, Italy. [10]Department of Pathology, University of Texas, MD Anderson Cancer Center, Houston, TX 77030, USA. [11]Department of Pathology, GZA-ZNA Ziekenhuizen, Antwerp, Belgium. [12]Division of Research, Peter MacCallum Cancer Centre, Melbourne, VIC, Australia. [13]Breast Medical Oncology, Yale Cancer Center, Yale School of Medicine, New Haven, CT, USA. [14]Division of Hematology-Oncology, Department of Medicine, School of Medicine, University of North Carolina at Chapel Hill, Chapel Hill, NC, USA. ✉e-mail: christos.sotiriou@hubruxelles.be

