## [Peer Review File · Nature Communications]

REVIEWER COMMENTS

Reviewer #1 (expertise in biostatistics):

In this manuscript, based on two phase III trials, NeoALTTO and CALGB 40601, the immune microenvironment (BCR and TCR) is identified the prognostic drivers for treatment optimization in HER2-positive breast cancer. The statistical analysis methods used in this manuscript are adaptable.

In general, the treatment response (pCR) and the event-free survival (EFS) are high correlated. Although the EFS affects the patient overall survival, the results based on the analysis of overall survival will be more significant.

Reviewer #2 (expertise in repertoire sequencing and analysis):

The role of lymphocytes in the tumor microenvironment was extensively studied during the last two decades. The insights gained from these studies propelled the development of advanced immunotherapies aiming at the activation of the patient's immune system to attack and eradicate the tumor cells or target directly the tumor cells. The immunotherapies include among others monoclonal antibodies (mAb) with either antagonistic or agonistic activities. One such example is the mAb trastuzumab (Herceptin) that is used to treat HER-positive breast cancer patients. It can be given as a combination therapy together with chemotherapy drugs such as lapatinib. Trastuzumab targets tumor cells that express the receptor tyrosine-protein kinase erbB-2 (HER2) and slows down cell replication. Moreover, it was found that neoadjuvant treatment escalation approaches with dual anti-HER2 blockade have been proven effective in human epidermal growth factor receptor 2 (HER2)-positive early-stage breast cancer, leading to an increase in pathological complete response (pCR) rates.

In the manuscript entitled "Immune and clinicopathological features predict HER2-positive breast cancer prognosis in the neoadjuvant NeoALTTO and CALGB 40601 trials", Rediti et al. aimed at investigating the role tumor-infiltrating lymphocytes (TILs) in the TME of HER-positive breast cancer patients and specifically what TIL characteristics that can facilitate the prediction of both pCR and prognosis. To achieve this aim the authors focused on evaluating the characteristics of the BCR and TCR repertoires in the context of two phase-III clinical trials, namely NeoALTTO and CALGB 40601.

The authors used prospective cohorts from the above-mentioned clinical trials where RNA-Seq data was generated. At first, the authors describe the characteristics of the RNA-Seq cohorts in the NeoALTTO and CALGB 40601 trials while elaborating on their clinical characteristics (although some were described in previous publications).

Next, the authors describe the methodology by which the BCR and TCR information was recovered from the RNA-Seq data. They note that BCR/TCR data depth was directly influenced by the sequencing length in which the datasets were generated (50 vs. 100 bp). In this context, it was not fully clear to me from the methodology description how clonotyping was carried out (see specific comments below). Moreover, they use a theoretical example to emphasize different B-cell clonal architectures and how these are used to describe the B/T cell response.

Next, they explored the association of BCR/TCR repertoires with hormone receptor (HR) status, PAM50 subtypes, and TIL levels, in order to evaluate the heterogeneity of the immune response within HER2-positive breast cancer. They note that HR-negative is associated with higher BCR clone numbers and reads in both cohorts. Moreover, exploring repertoire measures according to the stratification of PAM50 subtypes facilitates providing insight regarding the association with pCR and EFS rates within the two cohorts.

In order to determine how and whether BCR/TCR repertoire can predict pCR and EFS rates, they performed univariable and multivariable analyses in both cohorts. The results from this analysis suggest a positive prognostic role of B cell clonal expansion, depicted by lower BCR evenness and higher Gini index.

Next, the author described the development and validation of a predictive model that is based on treatment response, clinicopathological characteristics, and immune-related features. First, they

evaluated the model to predict EFS in the NEOALTTO cohort. Following several training steps, they determined the optimal model and the variables to be included in the model. They found that the final model should include BCR evenness, TIL levels, pCR status (ypT0/is), clinical nodal status, and hormone receptor status. Scores obtained by applying the model were used to stratify the patients into tertiles and Kaplan-Meier analysis was performed to estimate EFS in each group. Then, they proceeded to validate the prognostic model by computing the HER2-EveNT score in the CALGB 40601 set and applying the pre-defined cutoffs from NeoALTTO cohort.

They further seek to explore the relationship of BCR/TCR characteristics, TILs, and prognosis with the tumor microenvironment composition in both cohorts, by applying the microenvironment cell populations (MCP)-counter tool, which allowed them to estimate the abundance scores of different immune and stromal cells. Notably, they found remarkable differences in both studies within the RD subgroup, where levels of different immune cells were significantly higher in the good prognosis group, highlighting the importance of immune variables in patients with RD.

Lastly, they wanted to better characterize the gene expression features associated with the HER2-EveNT score. To that end, they evaluated the correlation between the HER2-EveNT score as a continuous variable and hallmark biological processes scores computed with the gene set variation analysis tool (GSVA) in the two cohorts. By applying the GSVA tool in both cohorts they found that immune-related pathways showed the highest negative correlation with the HER2-EveNT score and suggest that these results describe the association of proliferation and HER2 downstream signaling with pCR. In addition, factors that were described previously as linked to immune evasion mechanisms and tumor progression were positively correlated with the risk score and, thus, worse prognosis.

Furthermore, differential gene expression analysis compares the good and the poor prognostic groups in both cohorts. Eventually, they found an immune-enriched phenotype for the good prognosis group and enrichment in biological processes associated with a more aggressive phenotype and immune evasion in the poor prognosis group.

The author adequately describes that their study presented several challenges and limitations including the differences in patient populations and study design.

Overall, the authors conclude that they demonstrated the heterogeneity of BCR repertoire measures and immune response within HER2-positive breast cancer, and the potential of BCR diversity as a prognostic biomarker.

While the manuscript is well written including a concise introduction, in some parts the description was overcomplicated and included multiple associations with many parameters. Thus, throughout the manuscript, there are several parts that needed to be read several times to understand the take-home message. Maybe it should be better to start each description with the take-home message (in terms of results) and derive from that the full description of the results.

One of the major questions I have is the applicability of this analysis in the real world. Eventually, is it feasible to apply a methodology at the clinic that will provide information in a reasonable time frame that will facilitate a personalized treatment for each patient? Or in other words, in an event that each sample can be subjected to such analysis (that includes RNA-Seq, and in-depth of data including all key parameters), can the results be used to optimize the treatment strategy? Moreover, can this methodology pipeline be applied to each patient in the clinical setup?

Some technical and conceptual comments:

1) What do the number of BCR and TCR reads mean? the number of reads represents the product of the PCR amplification and not necessarily the expression level, how does this translate to the propensity of lymphocytes to infiltrate the tumor?

2) Clonality – it would be helpful if the authors will elaborate on the method that they cluster

clones. I have some concerns regarding this process – during the development of B cells in the bone marrow, the VDJ recombination of the heavy chain occurs first. Following successful recombination that is evaluated by the usage of a surrogate light chain, the B cell multiplies prior to the light chain VJ recombination occurring. This results in a situation in that different B cells express the same heavy chain with different light chains. Thus, when using the BCR-CDR3s of the heavy and light chains as a proxy to determine the clonality, they may over-cluster the data. It will be beneficial to get the author's feedback about this issue.

3) Supp Table 1: why the Number of reads normalized (median) is a positive value (1,212)? From my understanding, the number of reads that map to the BCR is normalized as relative to the total reads in the sample. The way the number of reads was normalized is not clear to me.

4) Line 157: how was the trimming done? I would expect that the number of identified clones (based on CDR3) will go down if the length was trimmed in-silico – the question is if this in-silico trimming represents the experimental sequencing of 50 bp. How will the resulting change if the trimming was done from the 5' or the 3' ?

5) Lines 179-181: It is unclear to me how were the global matrices calculated? Is this a single value representing all of the features of the repertoire?

6) Line 184-185: May be redundant? – similar ideas are written in lines 171-175. Moreover, I would expect that a higher number of reads will be associated with a higher repertoire diversity. The main question I have regarding this is: how was the data "cleaned" thus, what were the actions taken in order to ensure that sequences are without error (due to amplification errors and sequencing errors)?

7) TIL Level correlation: As TIL levels were evaluated for all samples – why wasn't it included in the correlation matrix as presented in Supp. Fig. 4 and 5? This is important to see if the number of lymphocytes determines the repertoire space – more cells -> more sequences. If there is a strong correlation (at least with a number of BCR/TCR reads) we can understand that the association of the parameters (as HR) with the number of reads has biological significance and is not related to the quality of data obtained for each sample.

8) Row 196-200: It is stated that HR- had significantly higher BCR read counts and a number of clones compared to HR+ tumors in both NeoALTTO and CALGB 40601, suggesting higher levels of B cell infiltration. Could it be that these data are correlated with the tumor size (thus more cells collected) rather than their propensity to infiltrate the tumor?

9) Line 199: How was the number of clones calculated - the number of clones is determined by each of the chains (IGH/L/K for BCR and a/b for TCR) so how was the number of clones calculated for the BCR ? using only the IGH/TCRb or integration of all 2/3 chains?

10) Row 234-237: " Since no strong correlation between TIL levels and BCR/TCR features was found, the information brought by TILs quantification and BCR/TCR characterization may be complementary" – the author suggests that the lack of correlation between TILs and BCR/TCR characterization is complementary, should they consider that the way the analysis was carried out could affect this? For example, clonotyping can be carried out in different ways which affect the biological insights (over clonotyping / under clonotyping). Moreover, it could be that the data in some cases is partial thus, does not represent the full landscape of BCR/TCR – this point is expected as the RNA-Seq experiments were not aimed to analyze the immune repertoire as opposed to BCR-Seq/TCR-Seq that is targeted at the comprehensive sequencing of the immune receptors (for example <https://doi.org/10.3389/fimmu.2021.705381>)

11) BCR isotypes: an important immune repertoire feature is isotype distribution. The isotype distribution enables us to understand better what is the effector function capacity of the antibodies produced by these B cells. Is it possible to recover from the RNA-seq data information regarding the isotype distribution of TIL-B?

12) Line 274: as BCR evenness and TIL levels was noted to be negatively correlated – how does

this affect their usage in the final model?

13) Line 258-259: "Instead, lower correlations were noted with TIL levels, with the highest value belonging to the B cell lineage in both studies ($\rho = 0.54$ to 0.56)." – it is not clear to me what is the main message of this sentence...

14) Line 359 (and supp figure 18): How was B cell / T cell lineage determined? methods describe how was clonality determined but not B/T cell lineage

Reviewer #3 (expertise in clinical oncology, ER-positive breast cancer):

This manuscript by Rediti et al. describes the development of a prognostic score based on RNA expression features of core biopsy of HER2+ breast cancer from patients treated in the NeoALTTO trial, with validation in a second similar neoadjuvant trial, CALGB 40601. Besides the use of 2 independent datasets, the work has a number of strengths, including that these were prospective clinical trials. The paper addresses an important clinical question, has novel findings, and the extended analyses provide insights into disease biology and provide a rich dataset for future analysis. Several limitations, including the need for further validation are appropriately noted. A few issues are deserving of clarification or further consideration:

1 - From the outset, the study considers pCR rate as a clinical feature that differs between datasets. However, given that the study treatments were not identical this is not a pure intrinsic feature of the patient population.

2 - The model development process incorporates pCR information, and this is a key component of the overall score. While later analyses show that the score is prognostic in both patients with pCR (in the discovery set) and residual disease (also validating in CALGB), it's not entirely clear why and whether including pCR in the initial feature selection provides the most clinically useful score: A strength of the molecular analyses is that it is performed at baseline, and thus in principle could inform neoadjuvant therapy. However, pCR information is (obviously) not available at this time, and thus the score cannot be applied.

Following the delivery of neoadjuvant therapy and the availability of pCR information which guides SOC use of T-DM1 and is the basis for many investigational strategies to escalate/de-escalate, the key need is to distinguish patients in each response group who will or will not have a subsequent recurrence event. Because pCR information was used in the initial feature selection, it is not clear that the other features included in the score provide optimal prognostic information for either of the pCR/non-pCR groups.

3 - Patients treated in these trials received investigational regimens incorporating lapatinib, which is not used in clinical practice, and the trials did show differences in pCR rate between arms. While the supplementary analysis nicely demonstrates that the prognostic effect is present across treatment groups, does the same hold in pCR/non-pCR patients where there may be confounding between treatment effect and the score? This seems most relevant to demonstrate in the non-lapatinib group.

4 - A variety of exploratory analyses provide some insights into the good/poor risk groups. However (unless I missed it), another similarly interesting exploratory analysis would be to consider any features that further discriminate EFS outcomes in each prognostic group - eg. comparing the good risk groups with non-pCR who do/do not have EFS events; comparing the poor risk groups with non-pCR who do/do not have EFS events etc. Has this been explored?

5 - The discussion acknowledges the role of RCB as a prognostic determinant in patients with residual disease. While I understand that more detailed refinement of the score to incorporate RCB is likely beyond the scope of the current work, it would be very informative to have some preliminary data exploring the relationships between RCB and score, or discriminatory performance in RCB class.

5 - While the discussion of IO strategies acknowledges the failure of the IMpassion050 trial, it

would be appropriate to note the toxicities also observed as well as the current state of investigation for early HER2+ disease (eg focussed on ADCs and TKIs), since this is a general interest rather than breast or oncology specialty journal.

RESPONSE TO REVIEWERS' COMMENTS

We are extremely thankful for the consideration of our manuscript, and we are deeply grateful to the Reviewers for the time dedicated to evaluating our manuscript and the precious feedback provided. We tried to the best of our possibilities to address all Reviewers' comments, and we believe that integrating their suggestions significantly improved the overall quality of the manuscript.

Please find below a point-by-point reply, with detailed descriptions of what has been modified in the paper (all changes are tracked and highlighted in the manuscript file). General updates include Figure 5 and Supplementary Figure 24, which were improved for visualization purposes, and the addition of Supplementary Tables currently listed as 4-7, 9, 13, 14, 17, 18, 23, 24, 34, 35 (see File containing the legend for the Supplementary Files for their description, and track changes in Manuscript files). Some references present in the first version of the manuscript (currently listed as 12, 60, 65) have also been updated (full studies published).

Reviewer #1 (expertise in biostatistics):

In this manuscript, based on two phase III trials, NeoALTTO and CALGB 40601, the immune microenvironment (BCR and TCR) is identified the prognostic drivers for treatment optimization in HER2-positive breast cancer. The statistical analysis methods used in this manuscript are adaptable. In general, the treatment response (pCR) and the event-free survival (EFS) are high correlated. Although the EFS affects the patient overall survival, the results based on the analysis of overall survival will be more significant.

Reply: We thank the Reviewer for the comments and for pointing out the relationship between event-free survival (EFS) and overall survival (OS). The decision to focus on EFS was driven not only by the correlation of this outcome with OS, but also by the clinical relevance of such endpoint for patients in terms of subsequent therapies and, overall, treatment strategies. Also, OS events are more likely in patients with an EFS event, so EFS is a good surrogate of OS while presenting significantly more events, leading to better statistical power. This being stated, we agree with the Reviewer that showing OS results is relevant in this setting. In this regard, it has to be mentioned that neither NeoALTTO nor CALGB 40601 were powered to detect differences in terms of survival endpoints, and our findings need further validation as already highlighted in the **Discussion** section of the manuscript.

Following the Reviewer's suggestion, we added Kaplan-Meier plots for OS in both NeoALTTO and CALGB 40601 (**Supplementary Figures 17** and **18**, respectively, see panels below), depicting the effect on OS of the prognostic groups we identified with our model and the HER2-Event score. OS was defined as the time from randomization to death from any cause. The OS definition has also been added in **Methods – NeoALTTO and CALGB 40601 study designs and patient populations**.

Supplementary Figure 17: Overall survival outcomes based on the groups identified by the prognostic HER2-Event model in the NeoALTT0 dataset.

Supplementary Figure 18: Overall survival outcomes based on the groups identified by the prognostic HER2-Event model in the CALGB 40601 independent validation dataset.

In NeoALTTO, the difference observed in EFS was also statistically significant for OS in the whole population (**Supplementary Figure 17a**), while the same trend in terms of improved outcome for the “good prognosis” group (with low HER2-Event score), although not statistically significant, was observed when focusing on the “pCR” (ypT0/is) and “no pCR” subgroups (**Supplementary Figure 17b** and **C**, respectively). The lack of statistical significance for the two subgroups (pCR and no-pCR) could be related to the reduced statistical power of this analysis, but the effect on the whole population is still observed.

Similarly, in CALGB 40601 the difference between the two prognostic groups remained significant in the whole population (**Supplementary Figure 18**, this time showing statistical significance in the “pCR” and “no pCR” subgroups as well (**Supplementary Figures 18b** and **c**, respectively). Of note, the difference in terms of outcome was not observed in the pCR subgroup when evaluating EFS.

The manuscript has been adapted accordingly (see paragraph: **Results – Development and validation of a prognostic model based on treatment response, clinicopathological characteristics and immune-related features – lines 354-359**:

“Results for overall survival (OS) are shown in Supplementary Figures 17 and 18. In both studies, OS results had the same trend compared to EFS; however, in NeoALTTO a statistically significant difference was observed only in the whole population and not in the pCR/no-pCR subgroups, while in CALGB 40601 statistically significant differences were observed in the pCR/no-pCR subgroups as well.”)

Reviewer #2 (expertise in repertoire sequencing and analysis):

The role of lymphocytes in the tumor microenvironment was extensively studied during the last two decades. The insights gained from these studies propelled the development of advanced immunotherapies aiming at the activation of the patient's immune system to attack and eradicate the tumor cells or target directly the tumor cells. The immunotherapies include among others monoclonal antibodies (mAb) with either antagonistic or agonistic activities. One such example is the mAb trastuzumab (Herceptin) that is used to treat HER-positive breast cancer patients. It can be given as a combination therapy together with chemotherapy drugs such as lapatinib. Trastuzumab targets tumor cells that express the receptor tyrosine-protein kinase erbB-2 (HER2) and slows down cell replication. Moreover, it was found that neoadjuvant treatment escalation approaches with dual anti-HER2 blockade have been proven effective in human epidermal growth factor receptor 2 (HER2)-positive early-stage breast cancer, leading to an increase in pathological complete response (pCR) rates.

In the manuscript entitled "Immune and clinicopathological features predict HER2-positive breast cancer prognosis in the neoadjuvant NeoALTTO and CALGB 40601 trials", Rediti et al. aimed at investigating the role tumor-infiltrating lymphocytes (TILs) in the TME of HER-positive breast cancer patients and specifically what TIL characteristics that can facilitate the prediction of both pCR and prognosis. To achieve this aim the authors focused on evaluating the characteristics of the BCR and TCR repertoires in the context of two phase-III clinical trials, namely NeoALTTO and CALGB 40601. The authors used prospective cohorts from the above-mentioned clinical trials where RNA-Seq data was generated. At first, the authors describe the characteristics of the RNA-Seq cohorts in the NeoALTTO and CALGB 40601 trials while elaborating on their clinical characteristics (although some were described in previous publications).

Next, the authors describe the methodology by which the BCR and TCR information was recovered from the RNA-Seq data. They note that BCR/TCR data depth was directly influenced by the sequencing length in which the datasets were generated (50 vs. 100 bp). In this context, it was not fully clear to me from the methodology description how clonotyping was carried out (see specific comments below). Moreover, they use a theoretical example to emphasize different B-cell clonal architectures and how these are used to describe the B/T cell response.

Next, they explored the association of BCR/TCR repertoires with hormone receptor (HR) status, PAM50 subtypes, and TIL levels, in order to evaluate the heterogeneity of the immune response within HER2-positive breast cancer. They note that HR-negative is associated with higher BCR clone numbers and reads in both cohorts. Moreover, exploring repertoire measures according to the stratification of PAM50 subtypes facilitates providing insight regarding the association with pCR and EFS rates within the two cohorts.

In order to determine how and whether BCR/TCR repertoire can predict pCR and EFS rates, they performed univariable and multivariable analyses in both cohorts. The results from this analysis suggest a positive prognostic role of B cell clonal expansion, depicted by lower BCR evenness and higher Gini index.

Next, the author described the development and validation of a predictive model that is based on treatment response, clinicopathological characteristics, and immune-related features. First, they evaluated the model to predict EFS in the NEOALTTO cohort. Following several training steps, they determined the optimal model and the variables to be included in the model. They found that the final model should include BCR evenness, TIL levels, pCR status (ypT0/is), clinical nodal status, and hormone receptor status. Scores obtained by applying the model were used to stratify the patients into tertiles and Kaplan-Meier analysis was performed to estimate EFS in each group. Then, they proceeded to validate the prognostic model by computing the HER2-Event score in the CALGB 40601 set and applying the pre-defined cutoffs from NeoALTTO

cohort.

They further seek to explore the relationship of BCR/TCR characteristics, TILs, and prognosis with the tumor microenvironment composition in both cohorts, by applying the microenvironment cell populations (MCP)-counter tool, which allowed them to estimate the abundance scores of different immune and stromal cells. Notably, they found remarkable differences in both studies within the RD subgroup, where levels of different immune cells were significantly higher in the good prognosis group, highlighting the importance of immune variables in patients with RD.

Lastly, they wanted to better characterize the gene expression features associated with the HER2-EveNT score. To that end, they evaluated the correlation between the HER2-EveNT score as a continuous variable and hallmark biological processes scores computed with the gene set variation analysis tool (GSVA) in the two cohorts. By applying the GSVA tool in both cohorts they found that immune-related pathways showed the highest negative correlation with the HER2-EveNT score and suggest that these results describe the association of proliferation and HER2 downstream signaling with pCR. In addition, factors that were described previously as linked to immune evasion mechanisms and tumor progression were positively correlated with the risk score and, thus, worse prognosis.

Furthermore, differential gene expression analysis compares the good and the poor prognostic groups in both cohorts. Eventually, they found an immune-enriched phenotype for the good prognosis group and enrichment in biological processes associated with a more aggressive phenotype and immune evasion in the poor prognosis group.

The author adequately describes that their study presented several challenges and limitations including the differences in patient populations and study design.

Overall, the authors conclude that they demonstrated the heterogeneity of BCR repertoire measures and immune response within HER2-positive breast cancer, and the potential of BCR diversity as a prognostic biomarker.

While the manuscript is well written including a concise introduction, in some parts the description was overcomplicated and included multiple associations with many parameters. Thus, throughout the manuscript, there are several parts that needed to be read several times to understand the take-home message. Maybe it should be better to start each description with the take-home message (in terms of results) and derive from that the full description of the results.

Reply: We are thankful to the Reviewer for the insightful summary of our findings and these first suggestions. We opted for building the different paragraphs concerning the results first describing the intent of the analyses, followed by the details and the take-home message summarizing the most important results at the end of each paragraph. Following the Reviewer's advice, we expanded the first part of some sections we considered, indeed, to be dense for results, and opted for maintaining the overall structures of the paragraphs.

One of the major questions I have is the applicability of this analysis in the real world. Eventually, is it feasible to apply a methodology at the clinic that will provide information in a reasonable time frame that will facilitate a personalized treatment for each patient? Or in other words, in an event that each sample can be subjected to such analysis (that includes RNA-Seq, and in-depth of data including all key parameters), can the results be used to optimize the treatment strategy? Moreover, can this methodology pipeline be applied to each patient in the clinical setup?

Reply: We thank the Reviewer for raising this extremely important point. We are aware of the limitations of the methodology as it is now, particularly for the BCR repertoire, requiring RNA-Seq (with associated costs and time frame for sequencing/analyses). While this represents a challenge for the clinical implementation of the score, we believe that these findings, first of all, provide the basis for further interrogating the role of B cells in this disease (further specified in **Discussion, lines 619-622**:

“Despite these considerations, findings from NeoALTTO were confirmed in CALGB 40601, suggesting the robustness of the results, which first and foremost provide the rationale for interrogating more extensively the role of B cells in this disease”). Results concerning the HER2DX test (**DOI: 10.1016/j.ebiom.2021.103801**, including the IgG gene signature we also evaluated when testing the variables to include in the model) also suggest an impact of immune response, particularly B cell mediated, on prognosis.

A potential step could be represented by the use of specific tests aiming at isolating and sequencing specifically the B (and T) cell receptors from tumor biopsies (e.g., polymerase chain reaction-based kits, but not only, as also mentioned by the Reviewer in **comment number 10**, and as now added in the **Discussion, lines 604-607** *“In addition, RNA sequencing-based methods to describe BCR/TCR may provide a partial picture of the repertoires compared to more sensitive technologies for BCR/TCR sequencing”*). Examples of specific BCR/TCR methodologies are also available at <https://mixcr.com/mixcr/reference/overview-built-in-presets/>.

Nonetheless, in order to ensure the reproducibility of our results and to facilitate the retrospective validation of the HER2-EveNT score in cohorts with RNA-Seq data available, or even the potential implementation in prospective studies, provided the availability of RNA-Seq (and we agree with the Reviewer that the time frame to get sequencing results and analyses could represent a bottleneck), we facilitate reproducibility and applicability of the HER2-EveNT score reporting:

- the BCR evenness average and standard deviation derived from NeoALTTO, which can be used to rescale the BCR evenness computed in an external cohort (the strategy used to validate the findings in the CALGB 40601 study);
- all the coefficients for the variables used to compute the score in any cohort with the data available;
- the script used to compute the BCR/TCR measures from the MiXCR output, which will be made available upon publication (see **Code availability**) and has been available for Reviewers in the review process. Details concerning how BCR/TCR measures were calculated are also described in the **Methods**.

As mentioned in the **Discussion**, a throughout validation of these findings in additional cohorts and settings would further strengthen the results and, potentially, lead to further exploration of specific tests to characterize the BCR and even improve our score.

Some technical and conceptual comments:

1) What do the number of BCR and TCR reads mean? the number of reads represents the product of the PCR amplification and not necessarily the expression level, how does this translate to the propensity of lymphocytes to infiltrate the tumor?

Reply: The number of BCR and TCR reads represent the number of reads mapping to CDR3 regions, as determined by RNA-Seq, and the cDNA library was not enriched with BCR or TCR sequences. Thus, the number of reads mapping to BCR/TCR, relative to the total number of reads mapping to the transcriptome, should give the same level of information about the expression of BCR/TCR as the

expression quantification of any other gene with RNA-Seq. The standard MiXCR pipeline was used, following the specific workflow for bulk RNA-Seq data.

We thank the Reviewer for the question, and this has been further clarified in the manuscript (**Methods – BCR and TCR repertoire diversity measures – lines 764-767**):

“read count reflects the number of reads uniquely mapping to CDR3 regions of BCR or TCR clones, and was normalized (i.e., divided) by the total number of reads mapping to the transcriptome in each sample and multiplied by a factor of 1000”.

Indeed, we observed a positive significant correlation between the normalized number of reads mapping to BCR/TCR and tumor-infiltrating lymphocytes (TILs) (in NeoALTTO, previously reported in a separate figure as **Supplementary Figures 11a and b**, now incorporated in **Supplementary Figure 4** as suggested by the Reviewer in the **comment number 7**, $\rho = 0.57$ and 0.33 for BCR and TCR normalized read counts; in CALGB 40601, previously as **Supplementary Figures 11c and d**, now in **Supplementary Figure 5**, $\rho = 0.58$ and 0.51 for BCR and TCR normalized read counts).

Thus, the normalized read counts seem to be associated to immune infiltration in the tumor, and in line with what is described for gene expression signatures describing immune-related processes (see also **DOI: 10.1001/jamaoncol.2022.6288**). Similar moderate correlation levels were also observed for immune-deconvolution (MCP-counter) results with TILs, see **Supplementary Figure 21**.

As we mention in our manuscript, the information provided by TILs and gene expression data could be complementary, rather than fully overlapping. However, we recognize that technical aspects, discussed in the following replies, may also have an impact on the correlation with TILs, and we added this consideration in the **Discussion (lines 608-609)**.

2) Clonality – it would be helpful if the authors will elaborate on the method that they cluster clones. I have some concerns regarding this process – during the development of B cells in the bone marrow, the VDJ recombination of the heavy chain occurs first. Following successful recombination that is evaluated by the usage of a surrogate light chain, the B cell multiplies prior to the light chain VJ recombination occurring. This results in a situation in that different B cells express the same heavy chain with different light chains. Thus, when using the BCR-CDR3s of the heavy and light chains as a proxy to determine the clonality, they may over-cluster the data. It will be beneficial to get the author's feedback about this issue.

Reply: We thank the Reviewer for raising another important point. In the MiXCR pipeline (v3.0.13), each chain type is clustered separately, and clones assigned as described in **Methods – BCR and TCR repertoire diversity measures – lines 730-749**: briefly, clones were defined as having the same CDR3 nucleotide sequence and V(D)J genes, and we aimed to assign each clone to the specific BCR and TCR chain using the V, D, and J genes information, whenever possible given the information obtained with MiXCR. When not possible (e.g., discordant information for the V and J genes), the clone was considered BCR or TCR without specifying the chain; this situation was however observed in a handful of cases (in **Supplementary Tables 2 and 3** we report the “global” BCR and TCR measures, as well as the ones derived for each chain).

To be precise, the BCR/TCR measures were computed on the chains separately as well as on all clones from reads mapping to BCR/TCR (the “global” ones, which were then used for the subsequent analyses). This has been further clarified in **Methods – BCR and TCR repertoire diversity measures – lines 758-760**: *“Those measures were computed for each BCR/TCR chain separately, as well as “global” measures considering all clones belonging to either BCR or TCR.”*

As we can see in **Supplementary Figures 4 and 5** (discussed in the paragraph **Results - BCR and TCR repertoires in the NeoALTTO and CALGB 40601 trials – lines 173-182**), measures related to the whole repertoire and to each individual chain are, for the most part, highly/moderately correlated (the main exception being the length of the CDR3 region, which is expected to be different between heavy and light chains, as, for example, also described in **DOI: 10.1093/bioinformatics/btw526**).

3) Supp Table 1: why the Number of reads normalized (median) is a positive value (1,212)? From my understanding, the number of reads that map to the BCR is normalized as relative to the total reads in the sample. The way the number of reads was normalized is not clear to me.

Reply: We apologize that the computation related to the normalized number of reads was not clear. As mentioned in reply to **comment number 1** as well, we rephrased this in the **Methods – BCR and TCR repertoire diversity measures (lines 764-767)** as follows:

“read count reflects the number of reads uniquely mapping to CDR3 regions of BCR or TCR clones, and was normalized (i.e., divided) by the total number of reads mapping to the transcriptome in each sample and multiplied by a factor of 1000”.

4) Line 157: how was the trimming done? I would expect that the number of identified clones (based on CDR3) will go down if the length was trimmed in-silico – the question is if this in-silico trimming represents the experimental sequencing of 50 bp. How will the resulting change if the trimming was done from the 5' or the 3' ?

Reply: We are thankful to the Reviewer for pointing out the lack of clarity for this part. From **line 160** (previously line 157), we now refer to details in **Methods**, and explain this further in the **RNA sequencing data processing** paragraph (**lines 700-707**), as follows:

*“When evaluating the potential technical differences between NeoALTTO and CALGB 40601 in terms of number of normalized BCR and TCR reads and clones, we considered the difference in read length as potential cause of such differences, and explored whether and how BCR/TCR read/clone counts would be impacted in NeoALTTO by trimming reads from 100-bp to 50-bp (**Supplementary Table 1**). Since the RNA sequencing was paired-end, the in-silico trimming from 100-bp to 50-bp was done from the middle of the fragment, representing what would be obtained by sequencing at 50-bp.”*

Indeed, the number of clones for both BCR and TCR went down in NeoALTTO when trimming in-silico from 100-bp to 50-bp, as depicted in **Supplementary Table 1**. Trimming from the 5' or 3', instead of from the middle, would simulate a situation in which not only less bases are sequenced, but also fragments are shorter, thus leading to worse results.

5) Lines 179-181: It is unclear to me how were the global matrices calculated? Is this a single value representing all of the features of the repertoire?

Reply: Indeed, the “global” BCR/TCR metrics are computed from all reads mapping to BCR or TCR chains, with clones separated as previously described, and as mentioned in **Results – BCR and TCR repertoires in the NeoALTTO and CALGB 40601 trials – lines 183-186**. Thus, they can be interpreted as a summary of the characteristics of the BCR/TCR repertoire as a whole. Please note (as described in the **Methods - BCR and TCR repertoire diversity measures – lines 741-749**), that we tried to assign a clone to a chain whenever possible based on the V, D and J genes. However, as previously mentioned, for some clones this was not possible, thus the global BCR and TCR read and clone numbers are not necessarily the exact sum of IGH/IGK/IGL or TRA/TRB/TRD/TRG.

6) Line 184-185: May be redundant? – similar ideas are written in lines 171-175. Moreover, I would expect that a higher number of reads will be associated with a higher repertoire diversity. The main question I have regarding this is: how was the data “cleaned” thus, what were the actions taken in order to ensure that sequences are without error (due to amplification errors and sequencing errors)?

Reply: Regarding the first point made by the Reviewer, we think in this case that the information provided in the two sections mentioned by the Reviewer are complementary. Indeed, **lines 173-182** (previously 171-175, please note that the line numbers have been updated after reviewing some

sections) touch upon the correlation between chains and global BCR/TCR measures depicted in **Supplementary Figures 4 and 5** (this has been detailed in the answer to **comment number 2**). **Lines 187-188 and following lines** (previously 184-185) expand, instead, on the correlations between the different BCR/TCR measures, showed as well in **Supplementary Figures 4 and 5** (read counts, number of clones, Gini index, evenness, Gini-Simpson index, top and second top clone proportions), thus describing a different concept.

In addition, while we agree that a higher number of reads mapping to BCR/TCR is expected to be associated to higher number of clones detected (Spearman of ~ 0.8 for BCR/TCR normalized read counts and number of clones in both NeoALTTO and CALGB 40601), the correlation with some of the diversity measures is, in our opinion, not implicit, although in general they indeed reflect the expectation of the Reviewer. In fact, taking as example evenness and Gini index, while a significant correlation was still present with the normalized number of reads, the magnitude of this correlation varied depending on the diversity measure (Spearman ~ 0.75 and ~ -0.5 for BCR Gini index and evenness, respectively, in the two studies; lower correlation values were instead noted for TCR; showed in **Supplementary Figures 4 and 5**).

Moreover, the correlations with the number of clones are less strong (for instance, ~ -0.2 between BCR evenness and number of clones in NeoALTTO, not significant in CALGB 40601), as situations with similar number of clones can lead to populations which are more or less balanced in terms of frequency of each clone (as also exemplified in **Supplementary Figure 1**).

With regards to the data cleaning, we followed the MiXCR (v3.0.13) pipeline for bulk RNA-Seq data with default parameters, which already implements cleaning steps with the removal of reads which are not considered reliable (also described in **DOI: 10.1038/nbt.3979**).

We thank the Reviewer for raising this point, because it allows us to expand on this topic. MiXCR is considered a “conservative” tool in terms of BCR/TCR detection, for instance as compared to other tools such as TRUST/TRUST4 (**DOI: 10.1038/s41588-018-0339-x**; **DOI: 10.1038/s41592-021-01142-2**). While of course different tools may have stronger and weaker aspects compared to others, a higher specificity was claimed for MiXCR as compared to TRUST (discussed in **DOI: 10.1038/nbt.3979**), with less false positive CDR3 sequences identified, potentially due to the different approaches adopted for partial CDR3 sequences. The evaluation of the differences between the tools has led to further in-depth comparisons of the tools (**DOI: 10.1038/ng.3820**; **DOI: 10.1038/nbt.4294**; **DOI: 10.1038/nbt.4296**).

Of course, a thorough comparison of the different tools is outside the scope of the present manuscript, but for our analyses we opted, at the time, for the use of MiXCR. As the field of immune-oncology is rapidly evolving, we can reasonably think that the arsenal of tools and their precision will also improve in the next years, as more data will be collected and more benchmarking will be performed.

In the MiXCR pipeline, there are several “quality control” steps, including the alignment stage and clonotype assembly (details are available at <https://mixcr.com/mixcr/reference/overview-analysis-overview/>). In this case, as rightly pointed out by the Reviewer, potential sequencing errors are important to consider, and the alignment step is particularly important.

MiXCR requires the full CDR3 region to be covered (as explained at <https://mixcr.com/mixcr/guides/rnaseq/?h=oallowpartialalignments#align>). Due to the fragmented nature of RNA-Seq data, with reads which may only partially cover the CDR3 sequence, we allowed, as detailed in the MiXCR workflow, the rescue of incomplete alignments due to the absence of V or J CDR3 part (option: `OallowPartialAlignments=true`), and followed the recommendations of the MiXCR pipeline. In particular, this step allows to assemble alignments that only partially cover the CDR3 region, and builds contigs from several initial alignments in a manner protected from artificial diversity generation (**DOI: 10.1038/nbt.3979**). This has been further clarified in the manuscript in **Methods – BCR and TCR repertoire diversity measures – lines 726-729**:

“BCR/TCR repertoires were extracted from bulk RNA sequencing data using the MiXCR tool^{29,30} (v3.0.13) with default parameters, specifying the option “OallowPartialAlignments = true”, and including all immunoglobulin and TCR chains”.

and 732-737:

“The option “OallowPartialAlignments” allowed to rescue incomplete alignments related to the absence of V or J CDR3 part, as the tool builds contigs from different initial alignments while avoiding artificial diversity generation³⁰. Indeed, this enables to assemble alignments only partially covering the CDR3 region, which is a possible occurrence due to the random coverage of BCR/TCR segments by fragmented reads in RNA sequencing.”

MiXCR performs two types of filtering, one based on PCR error correction, and one based on clonotypes quality. Across the NeoALTTO samples, a median of 3.4% of the clonotypes were eliminated by PCR error correction, and 6.6% were dropped because of low clonotype quality. In terms of reads mapping to the CDR3 region, this corresponds to a median of 1.1% of the mapped reads corrected by PCR error correction, and 13.5% of the reads dropped along with their low-quality clones.

No additional steps were implemented in addition to the ones included in the MiXCR pipeline. While MiXCR may be even considered too conservative by some Authors, the results in line with what could be biologically expected (relationship with immune-deconvolution and TILs – see next answer and answer to **comment number 1**), as well as the strength given by the validation of the findings in two independent cohorts (despite the technical differences in RNA-Seq described in the manuscript) point, in our opinion, toward the reliability of these results.

It is worth mentioning that since the same tool was applied on all samples in the two cohorts, any potential systematic bias, if present, would be similarly present on all samples. Because of this, significant differences between groups of samples should be robust, even if the values calculated (e.g., number of clones) could be systematically estimated lower or higher than the absolute truth (say as would be obtained by direct BCR sequencing).

7) TIL Level correlation: As TIL levels were evaluated for all samples – why wasn’t it included in the correlation matrix as presented in Supp. Fig. 4 and 5? This is important to see if the number of lymphocytes determines the repertoire space – more cells -> more sequences. If there is a strong correlation (at least with a number of BCR/TCR reads) we can understand that the association of the parameters (as HR) with the number of reads has biological significance and is not related to the quality of data obtained for each sample.

Reply: We thank the Reviewer for raising an important point. Indeed, TIL levels were available for most of the patients with RNA-Seq data (233/254 in NeoALTTO, and 230/264 in CALGB 40601, as described in the paragraph **Results – Heterogeneity of BCR and TCR repertoires according to hormone receptor status, PAM50 subtypes and correlations with TIL levels – lines 238-240**). As mentioned in reply to the **comment number 1**, the positive correlations between BCR/TCR reads and TIL levels point toward the biological significance of the number of BCR/TCR reads, rather than a potential technical artifact related to the sample/sequencing quality, and is in line with what is observed for several immune signatures (e.g., DOI: [10.1001/jamaoncol.2022.6288](https://doi.org/10.1001/jamaoncol.2022.6288)).

Following the Reviewer’s suggestion, we opted to add the correlation levels between BCR/TCR (including the different chains) and TILs in **Supplementary Figures 4 and 5**, removing **Supplementary Figure 11** which previously reported the correlation between global BCR/TCR measures and TILs. In addition, we added the Spearman correlation and P values used to create **Supplementary Figures 4 and 5** in **Supplementary Tables 4-5**.

8) Row 196-200: It is stated that HR- had significantly higher BCR read counts and a number of clones compared to HR+ tumors in both NeoALTTO and CALGB 40601, suggesting higher levels of B cell

infiltration. Could it be that these data are correlated with the tumor size (thus more cells collected) rather than their propensity to infiltrate the tumor?

Reply: We thank the Reviewer for raising this hypothesis. The total number of reads mapping to transcriptome were not significantly different according to hormone receptor status neither in NeoALTTO nor in CALGB 40601 (Wilcoxon rank sum test, P = 0.783 in NeoALTTO, P = 0.615 in CALGB 40601). With regards to tumor size, no differences according to tumor size (T1-T2 vs \geq T3) were observed according to hormone receptor status (Fisher's test, P = 0.608 in NeoALTTO, P = 0.768 in CALGB 40601), although it is worth mentioning that samples included in the analyses were all from baseline tumor biopsies.

The results obtained point toward higher B cell infiltration in HR- tumors (higher normalized number of reads mapping to BCR in HR-), a finding which would be in line with the higher immunogenicity attributed to HR- tumors (reviewed in DOI: [10.1186/s40425-019-0548-6](https://doi.org/10.1186/s40425-019-0548-6)). This last observation has been explicitly mentioned in the manuscript, to improve the clarity of the message (**Results – Heterogeneity of BCR and TCR repertoires according to hormone receptor status, PAM50 subtypes and correlations with TIL levels – lines 208-212:**

“In particular, HR- tumors had significantly higher BCR read counts and number of clones compared to HR+ tumors in both NeoALTTO and CALGB 40601, suggesting higher levels of B cell infiltration in line with a higher immunogenicity attributed to HR- tumors”.

9) Line 199: How was the number of clones calculated - the number of clones is determined by each of the chains (IGH/L/K for BCR and a/b for TCR) so how was the number of clones calculated for the BCR ? using only the IgH/TCRb or integration of all 2/3 chains?

Reply: We are thankful to the Reviewer for the opportunity to further clarify this aspect. As now better specified in the manuscript (**Methods – BCR and TCR repertoire diversity measures – lines 768-769**), *“the number of clones is defined as the total number of different clones in the sample”*, with clones defined as described in the same paragraph – **lines 730-749**.

As for other measures, the number of clones was computed for each chain and as global measure, summing all BCR (IGH, IGK, IGL) or TCR (TRA, TRB, TRD, TRG) chains.

Indeed, as previously mentioned in reply to the **comment number 2** and in the **Methods**, for some clones the assignment to a specific chain was not possible, thus the global BCR and TCR read and clone numbers are not necessarily the exact sum of IGH/IGK/IGL or TRA/TRB/TRD/TRG.

10) Row 234-237: “ Since no strong correlation between TIL levels and BCR/TCR features was found, the information brought by TILs quantification and BCR/TCR characterization may be complementary” – the author suggests that the lack of correlation between TILs and BCR/TCR characterization is complementary, should they consider that the way the analysis was carried out could affect this? For example, clonotyping can be carried out in different ways which affect the biological insights (over clonotyping / under clonotyping). Moreover, it could be that the data in some cases is partial thus, does not represent the full landscape of BCR/TCR – this point is expected as the RNA-Seq experiments were not aimed to analyze the immune repertoire as opposed to BCR-Seq/TCR-Seq that is targeted at the comprehensive sequencing of the immune receptors (for example <https://doi.org/10.3389/fimmu.2021.705381>)

Reply: We fully agree with the Reviewer that bulk RNA-Seq data derived BCR/TCR most likely leads to a reconstruction of the BCR/TCR repertoire that is not comprehensive (due to intrinsic limitations of the technology which is not targeted for BCR/TCR), and other more sensitive technologies could improve the characterization we provide (and, as mentioned in reply to the potential clinical implementation, even facilitate that aspect). We consider this a fair consideration, and we added this aspect in the **Discussion (lines 600-607)**.

We discussed about the correlation between TILs and BCR/TCR read counts in **comment number 1** and **number 7**. The correlation of TILs and BCR/TCR normalized number of reads is moderate, but in line with what we can observe when correlating MCP-counter results and TILs. Technical aspects, including potentially the use of different tools for clonotyping can potentially lead to slightly different results (although a comparison of the different tools is out of the scope of the present manuscript).

As we previously mentioned in reply to **comment number 6**, the choice of MiXCR was motivated also by the search for specificity in the BCR/TCR assembly. While this could potentially lead to under-estimation of clones present, and thus could potentially explain the moderate correlations observed, these correlations are in the same range of those for immune-related gene expression signatures (**DOI: 10.1001/jamaoncol.2022.6288**) or MCP-counter results (**Supplementary Figure 21**).

What we argue, and we thank the Reviewer for allowing us to clarify this aspect, is that higher TIL levels are not necessarily highly correlated to, for instance, evenness or Gini index, thus with measures of clonal expansion. Of course, a certain level of correlation between BCR/TCR features and TILs is still observed, as higher levels of TILs still point toward more immunogenic tumors (as also suggested by the direction of the correlations, for instance with BCR evenness and Gini index, **Supplementary Figures 4 and 5**). However, TILs may not be enough to describe if the immune response will also be fully effective, and if we are observing clonal expansion. The fact that both TILs and BCR evenness were selected in the prognostic model further strengthen this hypothesis, in our opinion.

We further clarified this point in the manuscript (**Results – Heterogeneity of BCR and TCR repertoires according to hormone receptor status, PAM50 subtypes and correlations with TIL levels – lines 249-254**), explicitly mentioning that BCR/TCR diversity measures may provide a “quality” measure of the immune response quantified by TILs:

“Since correlations between TIL levels and BCR/TCR diversity measures were moderate, the information brought by TILs quantification and BCR/TCR characterization may be complementary, with diversity measures potentially providing a “qualitative” information in terms of immune response compared to the quantification of TILs.”

11) BCR isotypes: an important immune repertoire feature is isotype distribution. The isotype distribution enables us to understand better what is the effector function capacity of the antibodies produced by these B cells. Is it possible to recover from the RNA-seq data information regarding the isotype distribution of TIL-B?

Reply: It is indeed possible to get the isotype from the RNA-Seq data for most clones, based on the information obtained from the constant region (C region). To address the Reviewer’s comment, for which we are thankful, we performed additional analyses.

Among the heavy chain (IGH) clones identified in the whole NeoALTTO cohort, for 23% we could not attribute the isotype (lack of constant region information). The remaining IGH were 55% IgG, 16% IgA, 5% IgM and <1% IgD and IgE. Similar distributions were observed in CALGB 40601: not determined isotype in 19% of the IGH clones, IgG 57%, IgA 20%, 4% IgM, <1% IgD and IgE. Due to the low amount of IgD and IgE clones identified, we focused the following analyses on the IgG, IgM and IgA isotypes.

We now define isotypes in **Methods – BCR and TCR repertoire and diversity measures – lines 750-754** (*“Isotypes (IgG, IgA, IgM, IgD, IgE) for the BCR heavy chain were determined based on the constant region (C region) information, whenever present. In particular, the isotype could not be determined in 23% of the IGH clones in NeoALTTO and 19% in CALGB 40601. For the remaining IGH clones, isotype distributions are shown in Supplementary Figures 3e-f”*).

Supplementary Figure 3 (below) has been modified to include **panels e and f** with the proportions of the isotypes (proportions were calculated after excluding BCR clones for which chain and/or isotype information could not be computed)

Supplementary Figure 3: Proportions of reads mapping to immunoglobulin chains/isotypes, and TCR chains in NeoALTTO and CALGB 40601.

We evaluated the correlation between the most relevant features identified in our analyses (global BCR normalized read counts, number of clones, evenness and Gini index) and those specific of the isotypes (thus computing those measures on the isotypes, following the same methodology described in the manuscript), to identify possible discordances in correlations for the different isotypes.

We added a paragraph in the **Results – BCR and TCR repertoires in the NeoALTTO and CALGB 40601 trials – lines 193-199:**

*“Isotypes (IgG, IgA, IgM, IgD, IgE) from BCR heavy chains were also computed (details in METHODS). In both studies, the majority of the BCR heavy chain reads were from IgG, followed by IgA and IgM (Supplementary Figure 3e-f). As shown in **Supplementary Figure 6**, positive correlations (the highest being for IgG) were observed between selected isotypes diversity measures and the corresponding global BCR measures.”*

We added **Supplementary Figure 6** (below) reporting the correlations of selected measures of IgG, IgM, IgA (normalized reads, number of clones, evenness and Gini index) with global BCR measures.

think that matching the B cell subtype with the isotype information goes beyond the possibilities for our analyses due to the already discussed technical limitations of bulk RNA-Seq.

12) Line 274: as BCR evenness and TIL levels was noted to be negatively correlated – how does this affect their usage in the final model?

Reply: Indeed, the coefficients in the model and score of TILs and BCR evenness are in line with their reciprocal negative correlation (see **Methods – Development of the prognostic model – lines 838-841**):

“The score was therefore calculated as:

BCR evenness (value) × 0.300436587 + TILs (%) × -0.024107504 + pCR (0 or 1) × -0.568111234 + HR status (0 or 1) × -0.525011259 + nodal status (0 or 1) × -0.504914127.”

This is shown also in **Supplementary Table 12 – rows 233-241**): negative coefficient for TILs and positive for BCR evenness (the direction is related to the fact that a lower score was associated with a better outcome, thus all variables correlated with good prognosis have a negative coefficient).

13) Line 258-259: “Instead, lower correlations were noted with TIL levels, with the highest value belonging to the B cell lineage in both studies (rho = 0.54 to 0.56).” – it is not clear to me what is the main message of this sentence...

Reply: In this case, we observed that BCR/TCR reads fit more with the results from MCP-counter than the ones from TILs (as also discussed in reply to **comment number 1** and **number 10**). This may be expected as RNA-Seq is the source for both MCP-counter and BCR/TCR analyses, although the information is taken from different genes/regions of the transcriptome (and, in our opinion, this can be considered reassuring in terms of reliability of the alignment process for BCR/TCR computation). We rephrased that section aiming at improving the clarity of the message in **Results – Characterization of the tumor microenvironment according to BCR/TCR repertoires and prognostic score – lines 392-399**:

“Instead, lower correlations between MCP-counter results and TIL levels were noted, with the highest value belonging to the B cell lineage in both studies (rho = 0.54 and 0.56 in NeoALTTO and CALGB 40601, respectively). These results are reassuring in that BCR/TCR measures highly correlate with B/T cells measured by MCP-counter, which may be expected as both are derived from RNA sequencing data, and suggest that MCP-counter fit better with BCR/TCR normalized reads than with TILs.”

14) Line 359 (and supp figure 18): How was B cell / T cell lineage determined? methods describe how was clonality determined but not B/T cell lineage

Reply: We apologize for the lack of clarity. In this case, the term “lineage” is the term adopted by the Authors of the MCP-counter paper (**DOI: 10.1186/s13059-016-1070-5**), and it incorporates all the B cells estimated by their tool which are scored together in one category, that is “B cell lineage”. Indeed, results described in **Characterization of the tumor microenvironment according to BCR/TCR repertoires and prognostic score** refer to the MCP-counter tool, as specified at the beginning of the paragraph (please see also **Methods - Molecular characterization of the prognostic groups: immune-deconvolution, gene set enrichment analysis, differential expression analysis, gene ontology analysis**, in which we renamed “B lymphocytes” to “B cell lineage” for consistency).

Reviewer #3 (expertise in clinical oncology, ER-positive breast cancer):

This manuscript by Rediti et al. describes the development of a prognostic score based on RNA expression features of core biopsy of HER2+ breast cancer from patients treated in the NeoALTTO trial, with validation in a second similar neoadjuvant trial, CALGB 40601. Besides the use of 2 independent datasets, the work has a number of strengths, including that these were prospective clinical trials. The paper addresses an important clinical question, has novel findings, and the extended analyses provide insights into disease biology and provide a rich dataset for future analysis. Several limitations, including the need for further validation are appropriately noted. A few issues are deserving of clarification or further consideration:

1 - From the outset, the study considers pCR rate as a clinical feature that differs between datasets. However, given that the study treatments were not identical this is not a pure intrinsic feature of the patient population.

Reply: The Reviewer raises a good point. We fully agree that the pCR rates are not intrinsic features of the datasets. The comparison (**Table 1** and mentioned in **Results – Characteristics of the RNA sequencing cohorts in the NeoALTTO and CALGB 40601 trials**) serves more the purpose of showing that pCR (the primary endpoint of the studies) was also different, but we agree that formally it makes sense to just report it as differences in rates of pCR and not include it in the “more favorable clinical features”. We thank the Reviewer for pointing this out, and we modified the paragraph accordingly (**line 130-134**: *“In particular, the CALGB 40601 cohort included a population characterized by more favorable clinical features (fewer \geq T3 tumors, more patients with clinically negative lymph nodes, higher TIL levels) compared to the NeoALTTO one. Higher pCR rates were also observed in CALGB 40601 compared to NeoALTTO (Table 1).”*), as well as the title of **Table 1**: *“Baseline patients’ characteristics and pathological complete response rates of the NeoALTTO and CALGB 40601 RNA sequencing cohorts.”* (maintaining the pCR information as we think that it is a useful piece of information).

2 - The model development process incorporates pCR information, and this is a key component of the overall score. While later analyses show that the score is prognostic in both patients with pCR (in the discovery set) and residual disease (also validating in CALGB), it’s not entirely clear why and whether including pCR in the initial feature selection provides the most clinically useful score: A strength of the molecular analyses is that it is performed at baseline, and thus in principle could inform neoadjuvant therapy. However, pCR information is (obviously) not available at this time, and thus the score cannot be applied.

Following the delivery of neoadjuvant therapy and the availability of pCR information which guides SOC use of T-DM1 and is the basis for many investigational strategies to escalate/de-escalate, the key need is to distinguish patients in each response group who will or will not have a subsequent recurrence event. Because pCR information was used in the initial feature selection, it is not clear that the other features included in the score provide optimal prognostic information for either of the pCR/non-pCR groups.

Reply: We thank the Reviewer for the comment and for allowing us to expand on this aspect. The main aim of that part of the analyses was to develop a prognostic model/score for EFS specific for the neoadjuvant setting. The fact that pCR was picked was indeed not unexpected, considering the known effect on prognosis.

It has to be noted that, in some cases, the prognosis could still be determined at baseline, without knowing the pCR information. In fact, this is the case for the patients with good prognosis and no-pCR, as well as those in the poor prognosis category for which achieving pCR would not change the prognostic category (although, for this subset EFS findings in NeoALTTO were not confirmed in

CALGB 40601, but in the latter a difference could still be observed for OS – see **additional analysis** suggested by **Reviewer #1** and **Supplementary Figure 18** added).

For instance, in the case of favorable features (e.g., node negative tumor, hormone receptor positive, high TILs, low evenness) the lack of pCR will not have any impact on the prognostic category, if the score obtained from the remaining variables is already within the threshold for “good prognosis”.

In this regard, as mentioned in **Results – Development and validation of an integrated prognostic model – lines 372-375**, we did not observe significant differences when comparing EFS in patients in the good prognosis group with or without pCR, although hazard ratios suggest a trend in favor of the pCR subgroup.

In addition, these findings lead us to think that in the absence of pCR, the other variables will play an even larger role. Indeed, one of the reasons why we pursued detailed analyses in the pCR and residual disease (RD) subgroups, comparing the prognostic categories and trying to further explore and understand the biological processes associated to the outcome, particularly in the RD subgroup, was also related to this point. As highlighted in **Supplementary Figures 22** and **25**, and as discussed in **Results – Characterization of the tumor microenvironment according to BCR/TCR repertoires and prognostic score – mostly lines 412-416** as well as in **Results – Gene expression profiling of the prognostic groups – mostly lines 485-490**, immune-related features, in particular, may become extremely important in determining the prognosis within the RD subgroup. This is again expected, as the model relies on immune-related features such as TILs and BCR evenness.

The inclusion of pCR in our model is also one of the reasons why, in the **Discussion**, we point toward post-treatment strategies, which may be modulated according to the score (at that point, influenced also by the pCR status).

To expand on the point made by the Reviewer, we believe some aspects are worth being considered:

- pCR is influenced also by non-biological characteristics, such as the treatment received (as shown by NeoALTTO and CALGB 40601 trials, as well as studies implementing the combination of pertuzumab and trastuzumab), and may not be, still, fully captured by baseline biomarkers, which would be needed to “replace” pCR in the model. Indeed, no tool is yet able to identify with 100% accuracy patients who will achieve pCR, although progresses are clearly being made, for instance with the HER2DX pCR score (**DOI: 10.1016/j.ebiom.2021.103801**).

If pCR was to be removed by the pool of variables, other non-biological variables may be picked instead. In fact, when testing this following Reviewer’s comment, the information about the treatment, that is whether the patient received the combination of trastuzumab and lapatinib, was selected, likely because a better treatment would lead to a higher chance of pCR.

Removing the treatment information as well (as trastuzumab + lapatinib does not represent standard of care) and following the same procedure described in the manuscript, ultimately led to the selection of TILs, BCR evenness, nodal status and ER status as variables, with a reduction of the performance of the prognostic model both in NeoALTTO (C-index 0.679 vs. 0.697 of the original model) and CALGB 40601 (C-index 0.6 vs. 0.639). Importantly, in CALGB 40601 we also lose the ability to detect differences between the two prognostic groups in patients with no pCR (log-rank P value = 0.19). This makes us think that including the pCR information could actually impact the weights of other variables picked, as in the original model including pCR they would have to outweigh the lack of pCR (which is a known information in the model) in terms of prognosis in the subgroup of patients with residual disease;

- the neoadjuvant setting presents some peculiar challenges in terms of biomarker discovery, as the relationship between prognostic/predictive biomarkers may be complex. Indeed, as for example shown by Fernandez-Martinez A, *et al.* (**DOI: 10.1200/JCO.20.01276**) and Prat A, *et al.* (**DOI: 10.1016/j.ebiom.2021.103801**), some features such as proliferation, HER2 signaling or luminal phenotype may provide discordant information for pCR and survival (e.g., for proliferation, higher

pCR rates, but also worse survival, and the opposite for luminal features). In the case of our model, for example, hormone receptor-positive tumors may not achieve pCR, but present good outcome.

In summary, we agree with the Reviewer that a model without the pCR information would allow us to evaluate the prognosis directly at baseline, while in this case a large portion of the score will be computed at baseline, with the pCR added at the end of the neoadjuvant phase. However, in the neoadjuvant setting, it is somehow complex to split the pCR and prognostic information, and the interactions of biomarkers are complex and pCR achievement, an important prognostic indicator, is difficult to recapitulate using only baseline features.

Despite this, patients with favorable prognostic features for the remaining variables, determining a score which is within the “good prognosis” threshold, may or may not achieve pCR and still present good outcome, thus allowing, at least for a subset of patients, to estimate the prognosis at baseline. This has been further clarified in the **Discussion – lines 550-557**:

“Based on our data, a combination of biomarkers performed on pre-treatment specimens may help identify a subset of patients (which according to our model may account for 20-30% of the patients with RD) with excellent outcomes and for whom post-operative trastuzumab alone could represent a viable option. Importantly, these patients can be identified computing the score with only the baseline features, as not achieving pCR will not have an impact on the prognostic category based on HER2-EveNT.”

3 - Patients treated in these trials received investigational regimens incorporating lapatinib, which is not used in clinical practice, and the trials did show differences in pCR rate between arms. While the supplementary analysis nicely demonstrates that the prognostic effect is present across treatment groups, does the same hold in pCR/non-pCR patients where there may be confounding between treatment effect and the score? This seems most relevant to demonstrate in the non-lapatinib group.

Reply: Indeed, the prognostic value of the identified groups was maintained across treatment arms (**Supplementary Figure 19**). We agree with the Reviewer that lapatinib, not being currently included in standard-of-care regimens in early-stage HER2-positive breast cancer, is less relevant (although a recent meta-analysis of 4 trials evaluating trastuzumab +/- lapatinib seems to suggest a potential role of the combination, due to an effect on survival; DOI: 10.1016/j.esmoop.2022.100433).

It is reasonable to expect that, by treatment arm and pCR status, the findings should be similar to what is shown in **Figures 3b and c** and **Figures 4b and c** (differences in prognosis by pCR status), and, as pointed out by the Reviewer, it could be worth verifying this in each arm. However, the number of patients for each prognostic group included in these six sub-analyses (pCR – defined as ypT0/is – trastuzumab, non-pCR trastuzumab, pCR combination, non-pCR combination, pCR lapatinib, non-pCR lapatinib) is quite low, leaving it difficult to find statistically significant differences, and, even if significant, the differences would be difficult to interpret in one sense or the other. In fact, as shown in the following **panel** (EFS curves comparing the prognostic groups according to pCR status and treatment arm for NeoALTTO – **a to f** – and CALGB 40601 – **g to l**), while all trends described in the whole population of the respective studies were maintained (please refer to **Figures 3b and c** and **Figure 4b and c** in the manuscript), only few comparisons were significant (to be precise, in NeoALTTO, the good vs. poor prognosis in pCR – lapatinib, log-rank P = 0.039; good vs. poor prognosis in pCR – trastuzumab, log-rank P = 0.016). Please also note that in CALGB 40601 the lapatinib arm was closed early.

The following panel has been added as **Supplementary Figure 20**.

Supplementary Figure 20: Event-free survival outcomes based on the groups identified by the prognostic model according to pCR status (ypT0/is) and treatment arm in the NeoALTT0 and CALGB 40601 studies.

In consideration of the small number of patients included in these comparisons, we think these results must be interpreted cautiously. This has been added in **Results – Development and validation of a prognostic model based on treatment response, clinicopathological characteristics and immune-related features – lines 364-371:**

“...and the difference in EFS outcome between the two prognostic groups could be observed across all treatment arms (Supplementary Figure 19). Furthermore, in each treatment arm the prognostic information associated to the HER2-EveNT groups in the pCR/no-pCR subsets seems to maintain the same trends observed in the whole cohort (Supplementary Figure 20), although these results must be interpreted with caution due to the small number of patients in each one of these comparisons”.

In addition, following the point made by the Reviewer, we now mention in the **Discussion** that the validation of these findings external cohorts is of value, particularly in those using standard treatment regimens (e.g., pertuzumab and trastuzumab) (**Discussion – lines 593-595**).

4 - A variety of exploratory analyses provide some insights into the good/poor risk groups. However (unless I missed it), another similarly interesting exploratory analysis would be to consider any features that further discriminate EFS outcomes in each prognostic group - eg. comparing the good risk groups with non-pCR who do/do not have EFS events; comparing the poor risk groups with non-pCR who do/do not have EFS events etc. Has this been explored?

Reply: We thank the Reviewer for the interesting point. However, this analysis is limited by the small number of events observed in each subset.

In NeoALTTO, 5 events were recorded in pCR (ypT0/is ypN0) + good prognosis, 13 in pCR + poor prognosis, 5 in RD + good prognosis, and 40 in RD + poor prognosis.

In CALGB 40601, 5 events were recorded in pCR + good prognosis, 4 in pCR + poor prognosis, 4 in RD + good prognosis, and 27 in RD + poor prognosis.

As the subgroup with more events and patients was the one with RD and poor prognosis, we conducted an analysis similarly to what we have done for **Supplementary Figure 25** (described in **Results – Gene expression profiling of the prognostic groups – lines 474-484**), aiming at trying to address Reviewer’s comment and evaluating whether 98 relevant genes/gene expression signatures could differentiate tumors with/without event within this subset of patients.

However, such comparison did not show any significant differences when adjusting for multiple testing in NeoALTTO.

This finding is not unexpected, since the prognostic groups were already identified using the EFS information.

In CALGB 40601, the levels of only 2 signatures

(Sig_TCGA_BRCA_1198_TP63_JCI_2020_PMID_32573490,

Sig_TCGA_BRCA_1198_BASAL_JCI_2020_PMID_32573490) and 1 gene (MET) resulted significantly (FDR < 0.05) lower in patients with an event within the RD + poor prognosis group.

It has to be noted (as also discussed in the next answer to **comment number 5**) that this particular analysis would benefit greatly from the knowledge of the RCB, as it could represent a confounding factor for prognosis, and since the findings were based on a small subset of patients and not confirmed in NeoALTTO (where the direction of the association of the 2 signatures with “event” was the opposite, although not significant) we would refrain from speculating over the results obtained.

5 - The discussion acknowledges the role of RCB as a prognostic determinant in patients with residual disease. While I understand that more detailed refinement of the score to incorporate RCB is likely beyond the scope of the current work, it would be very informative to have some preliminary data exploring the relationships between RCB and score, or discriminatory performance in RCB class.

Reply: We thank the Reviewer for raising this important point. Unfortunately, the residual cancer burden (RCB) information was not available for the present analyses in the evaluated trials, but we agree, as indeed mentioned in the **Discussion**, that including RCB data could further improve the prognostic information, and we see the value of studying RCB classes to derive relevant biological information (in the manuscript we cited the work by Sammut SJ, et al. DOI: [10.1038/s41586-021-](https://doi.org/10.1038/s41586-021-)

04278-5, which evaluated biological processes associated to higher/lower RBC, among other analyses, when mentioning that neural processes, associated to poor prognosis, were also associated to increased RCB in that paper – **line 449-450**). We share the hope that more translational analyses will be performed in neoadjuvant trials with the RCB information available.

5 - While the discussion of IO strategies acknowledges the failure of the IMpassion050 trial, it would be appropriate to note the toxicities also observed as well as the current state of investigation for early HER2+ disease (eg focussed on ADCs and TKIs), since this is a general interest rather than breast or oncology specialty journal.

Reply: We thank the Reviewer for mentioning this, and we agree that this is a relevant point to raise in the **Discussion**. Therefore, we added the following sentences:

- “, and adverse events were more frequent in the atezolizumab arm” in **Discussion – lines 566-567**;
- “, considering that a precise selection of patients is crucial due to the potential associated toxicities”

Discussion – lines 571-572

We already mentioned the importance of selecting patients who may benefit from T-DM1, considering the associated toxicities, in **Discussion – lines 542-544**.

We also agree that it is worth mentioning that the treatment landscape of early-stage HER2-positive breast cancer is evolving and will likely change in the upcoming years. Although a detailed description of the potential strategies is outside the scope of the manuscript, we mention this concept in **Discussion – lines 576-579** and we added the recent reference **DOI: 10.1038/s41573-022-00579-0**, which in our opinion covers also this topic (“*Of note, the treatment arsenal of HER2-positive breast cancer is rapidly evolving, with several agents (e.g., antibody–drug conjugates and novel tyrosine kinase inhibitors) which are being evaluated in the early-stage setting⁶⁵ and may reshape the current biomarker landscape*”).

REVIEWERS' COMMENTS

Reviewer #1 (expertise in biostatistics):

This revised manuscript has greatly improved and all of my comments have been addressed. I have no further comments.

Reviewer #2 (expertise in repertoire sequencing and analysis):

The authors fully addressed all my concerns. They amended the manuscript to include additional analysis requested by myself and other reviewers and provided a comprehensive and detailed "point-by-point" response.

I thank the authors for investing all the time to improve the manuscript and I am happy to endorse the publication of this important study.

Reviewer #3 (expertise in clinical oncology, ER-positive breast cancer):

The authors have provided substantial responses to my previous comments, including additional data and analyses. Overall these have satisfied most of my main questions and improved the manuscript.

I have one followup comment, noted below in the context of the rebuttal letter thread:

2 - The model development process incorporates pCR information, and this is a key component of the overall score. While later analyses show that the score is prognostic in both patients with pCR (in the discovery set) and residual disease (also validating in CALGB), it's not entirely clear why and whether including pCR in the initial feature selection provides the most clinically useful score: A strength of the molecular analyses is that it is performed at baseline, and thus in principle could inform neoadjuvant therapy. However, pCR information is (obviously) not available at this time, and thus the score cannot be applied.

Following the delivery of neoadjuvant therapy and the availability of pCR information which guides SOC use of T-DM1 and is the basis for many investigational strategies to escalate/de-escalate, the key need is to distinguish patients in each response group who will or will not have a subsequent recurrence event. Because pCR information was used in the initial feature selection, it is not clear that the other features included in the score provide optimal prognostic information for either of the pCR/non-pCR groups.

Reply: We thank the Reviewer for the comment and for allowing us to expand on this aspect. The main aim of that part of the analyses was to develop a prognostic model/score for EFS specific for the neoadjuvant setting. The fact that pCR was picked was indeed not unexpected, considering the known effect on prognosis.

It has to be noted that, in some cases, the prognosis could still be determined at baseline, without knowing the pCR information. In fact, this is the case for the patients with good prognosis and no-pCR, as well as those in the poor prognosis category for which achieving pCR would not change the prognostic category (although, for this subset EFS findings in NeoALTTO were not confirmed in CALGB 40601, but in the latter a difference could still be observed for OS – see additional analysis suggested by Reviewer #1 and Supplementary Figure 18 added).

For instance, in the case of favorable features (e.g., node negative tumor, hormone receptor positive, high TILs, low evenness) the lack of pCR will not have any impact on the prognostic category, if the score obtained from the remaining variables is already within the threshold for "good prognosis".

In this regard, as mentioned in Results – Development and validation of an integrated prognostic model – lines 372-375, we did not observe significant differences when comparing EFS in patients in the good prognosis group with or without pCR, although hazard ratios suggest a trend in favor of the pCR subgroup.

In addition, these findings lead us to think that in the absence of pCR, the other variables will play an even larger role. Indeed, one of the reasons why we pursued detailed analyses in the pCR and

residual disease (RD) subgroups, comparing the prognostic categories and trying to further explore and understand the biological processes associated to the outcome, particularly in the RD subgroup, was also related to this point. As highlighted in Supplementary Figures 22 and 25, and as discussed in Results – Characterization of the tumor microenvironment according to BCR/TCR repertoires and prognostic score – mostly lines 412-416 as well as in Results – Gene expression profiling of the prognostic groups – mostly lines 485-490, immune-related features, in particular, may become extremely important in determining the prognosis within the RD subgroup. This is again expected, as the model relies on immune-related features such as TILs and BCR evenness. The inclusion of pCR in our model is also one of the reasons why, in the Discussion, we point toward post-treatment strategies, which may be modulated according to the score (at that point, influenced also by the pCR status).

To expand on the point made by the Reviewer, we believe some aspects are worth being considered:

- pCR is influenced also by non-biological characteristics, such as the treatment received (as shown by NeoALTTO and CALGB 40601 trials, as well as studies implementing the combination of pertuzumab and trastuzumab), and may not be, still, fully captured by baseline biomarkers, which would be needed to “replace” pCR in the model. Indeed, no tool is yet able to identify with 100% accuracy patients who will achieve pCR, although progresses are clearly being made, for instance with the HER2DX pCR score (DOI: 10.1016/j.ebiom.2021.103801).

If pCR was to be removed by the pool of variables, other non-biological variables may be picked instead. In fact, when testing this following Reviewer’s comment, the information about the treatment, that is whether the patient received the combination of trastuzumab and lapatinib, was selected, likely because a better treatment would lead to a higher chance of pCR.

Removing the treatment information as well (as trastuzumab + lapatinib does not represent standard of care) and following the same procedure described in the manuscript, ultimately led to the selection of TILs, BCR evenness, nodal status and ER status as variables, with a reduction of the performance of the prognostic model both in NeoALTTO (C-index 0.679 vs. 0.697 of the original model) and CALGB 40601 (C-index 0.6 vs. 0.639). Importantly, in CALGB 40601 we also lose the ability to detect differences between the two prognostic groups in patients with no pCR (log-rank P value = 0.19). This makes us think that including the pCR information could actually impact the weights of other variables picked, as in the original model including pCR they would have to outweigh the lack of pCR (which is a known information in the model) in terms of prognosis in the subgroup of patients with residual disease;

- the neoadjuvant setting presents some peculiar challenges in terms of biomarker discovery, as the relationship between prognostic/predictive biomarkers may be complex. Indeed, as for example shown by Fernandez-Martinez A, et al. (DOI: 10.1200/JCO.20.01276) and Prat A, et al. (DOI: 10.1016/j.ebiom.2021.103801), some features such as proliferation, HER2 signaling or luminal phenotype may provide discordant information for pCR and survival (e.g., for proliferation, higher pCR rates, but also worse survival, and the opposite for luminal features). In the case of our model, for example, hormone receptor-positive tumors may not achieve pCR, but present good outcome.

In summary, we agree with the Reviewer that a model without the pCR information would allow us to evaluate the prognosis directly at baseline, while in this case a large portion of the score will be computed at baseline, with the pCR added at the end of the neoadjuvant phase. However, in the neoadjuvant setting, it is somehow complex to split the pCR and prognostic information, and the interactions of biomarkers are complex and pCR achievement, an important prognostic indicator, is difficult to recapitulate using only baseline features.

Despite this, patients with favorable prognostic features for the remaining variables, determining a score which is within the “good prognosis” threshold, may or may not achieve pCR and still present good outcome, thus allowing, at least for a subset of patients, to estimate the prognosis at baseline.

This has been further clarified in the Discussion – lines 550-557:

“Based on our data, a combination of biomarkers performed on pre-treatment specimens may help identify a subset of patients (which according to our model may account for 20-30% of the patients with RD) with excellent outcomes and for whom post-operative trastuzumab alone could represent a viable option. Importantly, these patients can be identified computing the score with only the baseline features, as not achieving pCR will not have an impact on the prognostic

category based on HER2-Event.”

----- I appreciate the authors’ consideration of this important point. I’m not sure that this line added to the discussion captures the full complexity of the issue as they have elaborated above. It’s important to me that omitting pCR information from the model development did not result in a dramatically different model or impact the utility of other biological information that is the focus of this study. I defer to the statistician on those points. Ensuring that the pCR issues is addressed in other parts of the manuscript where it is relevant would be helpful (eg Line 286-91 where it is described as part of the “baseline” clinicopathologic information. (“Considering the previous results, we tested whether a model including baseline clinicopathological and immune-related features could predict EFS in NeoALTTO. BCR and TCR measures (details in METHODS) were included in the variable selection process, together with pCR information, clinicopathological variables and a set of gene signatures.”). And perhaps other areas.

RESPONSE TO REVIEWERS' COMMENTS

Reviewer #1 (expertise in biostatistics):

This revised manuscript has great improved and all of my comments have been addressed. I have no further comments.

Reviewer #2 (expertise in repertoire sequencing and analysis):

The authors fully addressed all my concerns. They amended the manuscript to include additional analysis requested by myself and other reviewers and provided a comprehensive and detailed "point-by-point" response.

I thank the authors for investing all the time to improve the manuscript and I am happy to endorse the publication of this important study.

Reply to Reviewers #1 and #2: We are deeply thankful for the Reviewers' #1 and #2 positive comments and for the time spent reviewing our manuscript and our replies. We believe their comments and insights provided the opportunity to improve the quality of the paper.

Reviewer #3 (expertise in clinical oncology, ER-positive breast cancer):

The authors have provided substantial responses to my previous comments, including additional data and analyses. Overall these have satisfied most of my main questions and improved the manuscript.

I have one followup comment, noted below in the context of the rebuttal letter thread:

2 - The model development process incorporates pCR information, and this is a key component of the overall score. While later analyses show that the score is prognostic in both patients with pCR (in the discovery set) and residual disease (also validating in CALGB), it's not entirely clear why and whether including pCR in the initial feature selection provides the most clinically useful score: A strength of the molecular analyses is that it is performed at baseline, and thus in principle could inform neoadjuvant therapy. However, pCR information is (obviously) not available at this time, and thus the score cannot be applied.

Following the delivery of neoadjuvant therapy and the availability of pCR information which guides SOC use of T-DM1 and is the basis for many investigational strategies to escalate/de-escalate, the key need is to distinguish patients in each response group who will or will not have a subsequent recurrence event. Because pCR information was used in the initial feature selection, it is not clear that the other features included in the score provide optimal prognostic information for either of the pCR/non-pCR groups.

Reply: We thank the Reviewer for the comment and for allowing us to expand on this aspect. The main aim of that part of the analyses was to develop a prognostic model/score for EFS specific for the neoadjuvant setting. The fact that pCR was picked was indeed not unexpected, considering the known effect on prognosis.

It has to be noted that, in some cases, the prognosis could still be determined at baseline, without knowing the pCR information. In fact, this is the case for the patients with good prognosis and no-pCR, as well as those in the poor prognosis category for which achieving pCR would not change the

prognostic category (although, for this subset EFS findings in NeoALTTO were not confirmed in CALGB 40601, but in the latter a difference could still be observed for OS – see additional analysis suggested by Reviewer #1 and Supplementary Figure 18 added).

For instance, in the case of favorable features (e.g., node negative tumor, hormone receptor positive, high TILs, low evenness) the lack of pCR will not have any impact on the prognostic category, if the score obtained from the remaining variables is already within the threshold for “good prognosis”.

In this regard, as mentioned in Results – Development and validation of an integrated prognostic model – lines 372-375, we did not observe significant differences when comparing EFS in patients in the good prognosis group with or without pCR, although hazard ratios suggest a trend in favor of the pCR subgroup.

In addition, these findings lead us to think that in the absence of pCR, the other variables will play an even larger role. Indeed, one of the reasons why we pursued detailed analyses in the pCR and residual disease (RD) subgroups, comparing the prognostic categories and trying to further explore and understand the biological processes associated to the outcome, particularly in the RD subgroup, was also related to this point. As highlighted in Supplementary Figures 22 and 25, and as discussed in Results – Characterization of the tumor microenvironment according to BCR/TCR repertoires and prognostic score – mostly lines 412-416 as well as in Results – Gene expression profiling of the prognostic groups – mostly lines 485-490, immune-related features, in particular, may become extremely important in determining the prognosis within the RD subgroup. This is again expected, as the model relies on immune-related features such as TILs and BCR evenness.

The inclusion of pCR in our model is also one of the reasons why, in the Discussion, we point toward post-treatment strategies, which may be modulated according to the score (at that point, influenced also by the pCR status).

To expand on the point made by the Reviewer, we believe some aspects are worth being considered:
- pCR is influenced also by non-biological characteristics, such as the treatment received (as shown by NeoALTTO and CALGB 40601 trials, as well as studies implementing the combination of pertuzumab and trastuzumab), and may not be, still, fully captured by baseline biomarkers, which would be needed to “replace” pCR in the model. Indeed, no tool is yet able to identify with 100% accuracy patients who will achieve pCR, although progresses are clearly being made, for instance with the HER2DX pCR score (DOI: 10.1016/j.ebiom.2021.103801).

If pCR was to be removed by the pool of variables, other non-biological variables may be picked instead. In fact, when testing this following Reviewer’s comment, the information about the treatment, that is whether the patient received the combination of trastuzumab and lapatinib, was selected, likely because a better treatment would lead to a higher chance of pCR.

Removing the treatment information as well (as trastuzumab + lapatinib does not represent standard of care) and following the same procedure described in the manuscript, ultimately led to the selection of TILs, BCR evenness, nodal status and ER status as variables, with a reduction of the performance of the prognostic model both in NeoALTTO (C-index 0.679 vs. 0.697 of the original model) and CALGB 40601 (C-index 0.6 vs. 0.639). Importantly, in CALGB 40601 we also lose the ability to detect differences between the two prognostic groups in patients with no pCR (log-rank P value = 0.19). This makes us think that including the pCR information could actually impact the weights of other variables picked, as in the original model including pCR they would have to outweigh the lack of pCR (which is a known information in the model) in terms of prognosis in the subgroup of patients with residual disease;

- the neoadjuvant setting presents some peculiar challenges in terms of biomarker discovery, as the relationship between prognostic/predictive biomarkers may be complex. Indeed, as for example shown by Fernandez-Martinez A, et al. (DOI: 10.1200/JCO.20.01276) and Prat A, et al. (DOI: 10.1016/j.ebiom.2021.103801), some features such as proliferation, HER2 signaling or luminal phenotype may provide discordant information for pCR and survival (e.g., for proliferation, higher

pCR rates, but also worse survival, and the opposite for luminal features). In the case of our model, for example, hormone receptor-positive tumors may not achieve pCR, but present good outcome.

In summary, we agree with the Reviewer that a model without the pCR information would allow us to evaluate the prognosis directly at baseline, while in this case a large portion of the score will be computed at baseline, with the pCR added at the end of the neoadjuvant phase. However, in the neoadjuvant setting, it is somehow complex to split the pCR and prognostic information, and the interactions of biomarkers are complex and pCR achievement, an important prognostic indicator, is difficult to recapitulate using only baseline features.

Despite this, patients with favorable prognostic features for the remaining variables, determining a score which is within the “good prognosis” threshold, may or may not achieve pCR and still present good outcome, thus allowing, at least for a subset of patients, to estimate the prognosis at baseline. This has been further clarified in the Discussion – lines 550-557:

“Based on our data, a combination of biomarkers performed on pre-treatment specimens may help identify a subset of patients (which according to our model may account for 20-30% of the patients with RD) with excellent outcomes and for whom post-operative trastuzumab alone could represent a viable option. Importantly, these patients can be identified computing the score with only the baseline features, as not achieving pCR will not have an impact on the prognostic category based on HER2-EveNT.”

---- I appreciate the authors’ consideration of this important point. I’m not sure that this line added to the discussion captures the full complexity of the issue as they have elaborated above. It’s important to me that omitting pCR information from the model development did not result in a dramatically different model or impact the utility of other biological information that is the focus of this study. I defer to the statistician on those points. Ensuring that the pCR issues is addressed in other parts of the manuscript where it is relevant would be helpful (eg Line 286-91 where it is described as part of the “baseline” clinicopathologic information. (“Considering the previous results, we tested whether a model including baseline clinicopathological and immune-related features could predict EFS in NeoALTTO. BCR and TCR measures (details in METHODS) were included in the variable selection process, together with pCR information, clinicopathological variables and a set of gene signatures.”). And perhaps other areas.

Reply: We thank the Reviewer for the comment and for the time spent reviewing our manuscript and our replies. We agree with the Reviewer’s follow up comment, and to address it we clarified the sentence mentioned by the Reviewer: *“Considering the previous results, we tested whether a model including baseline clinicopathological, immune-related features and treatment-related information including response could predict EFS in NeoALTTO...”* (paragraph **Results – Development and validation of an integrated prognostic model – lines 294-296**).

Moreover, in the same paragraph, we added the following section mentioning that similar variables were selected when excluding pCR and treatment information (**lines 386-391**):

“Similar variables (i.e., TILs, BCR evenness, nodal status, and estrogen receptor status) were selected when removing the pCR and treatment information from the pool of variables tested, although this version of the EFS model presented a reduced performance compared to the original one, both in NeoALTTO (C-index 0.6787 vs. 0.6979 of the original model) and CALGB 40601 (C-index 0.6029 vs. 0.6396).”

In the following sentence (**lines 392-394**), we mentioned “response information”:

“Overall, a multi-modal approach integrating clinicopathological characteristics, response information and immunological features is important to predict patients’ outcomes”.

Moreover, we added in the **Discussion** the following section, summarizing the main points addressed in our previous reply to the Reviewer’s comment (**lines 535-548**):

“While pCR achievement is a strong prognostic factor in HER2-positive breast cancers after neoadjuvant therapy^{27,28}, the selection of further variables in the model suggests the importance of additional processes/features in determining prognosis. In fact, the risk of relapse is influenced by the interaction of several variables, particularly in the absence of pCR where the other biomarkers play a crucial role. The neoadjuvant setting presents some peculiar challenges in terms of biomarker discovery, as the relationship between prognostic features (related to survival/relapse risk) and predictive markers (related to pCR) may be complex. Indeed, some features such as proliferation, HER2 signaling or luminal phenotype may provide discordant information for pCR and survival (e.g., lower pCR rates and improved outcome for luminal features)^{6,55}. As additional factors such as treatment administered could impact response to neoadjuvant therapies, pCR achievement may not be, still, fully captured by pre-treatment biomarkers, though progresses in this regard are being made⁵⁵.”

Finally, in the sentence at **lines 657-658** we mentioned “treatment response information”:
“...and highlighted the importance of integrating clinicopathological characteristics, treatment response information and immune-related features to define the clinical risk in the neoadjuvant setting”.